

# Barchan swarm dynamics from a Two-Flank Agent-Based Model

Dominic T Robson[1] and Andreas C W Baas[1]

[1]King's College London, Department of Geography,Bush House, North East Wing, 40 Aldwych, London, UK, WC2B 4BG

**Correspondence:** Dominic Robson (dominic.robson@kcl.ac.uk)

**Abstract.** We perform simulations of barchan swarms using the Two-Flank Agent-Based model investigating the effects of changing the angular separation between primary and secondary modes of wind, the density at which new dunes are injected, and the parameter $q_{shift}$ which controls the rate at which sediment is reorganised to restore symmetry in an asymmetric dune. Unlike previous agent-based models, we are able to produce longitudinally homogeneous size distributions and, for sparse swarms, steady longitudinal number density. We are able to constrain $q_{shift}$ by comparing the range of values for which longitudinally stability is observed with the range of values for which the width of asymmetry distributions is consistent with real-world swarms. Furthermore, we demonstrate dune size, asymmetry, dune density, spatial alignment, and collision dynamics are all strongly influenced by the angular separation of bimodal winds.

## 1 Introduction

Crescent-shaped barchan dunes are one of the most striking aeolian bedforms and are found on Mars (Bourke, 2010; Rubanenko et al., 2022; Bourke and Goudie, 2009; Robson et al., 2022) and many different locations on Earth (Goudie, 2020; Elbelrhiti et al., 2008; Hersen et al., 2004; Robson et al., 2022). Compared to other types of bedform, barchans migrate quickly, often up to hundreds of metres per year (Sparavigna, 2013; Corbett, 2018; Barnes, 2001). The rate at which the dunes move is controlled primarily by their size (Long and Sharp, 1964; Bagnold, 1941) which means that collisions can occur between the bedforms.

Individual barchans are typically tens of metres wide (Elbelrhiti et al., 2008; Durán et al., 2009) and evolve over timescales on the order of years. The evolution of small numbers of such bedforms can, therefore, be studied using numerical models of the microscopic behaviours of the sediment and fluid flow (Durán et al., 2011; Xiao et al., 2023). It is also possible to study proxies of aeolian barchans in a laboratory setting by using a fluid, e.g. water, which is much denser than air (Hersen, 2005; Hersen et al., 2002; Assis and Franklin, 2020, 2021). In nature, however, we do not typically observe barchans forming in small numbers but rather vast populations of interacting dunes (Hersen et al., 2004). These huge systems, known as swarms (Hesse, 2009; Robson et al., 2022), can span tens or hundreds of square kilometres and contain hundreds, thousands or even tens of thousands of bedforms (King, 1918; Durán et al., 2009; Elbelrhiti et al., 2008).

Not only are the length scales of swarms orders of magnitude larger than those of the dunes themselves, but the timescales over which these systems evolve are also much longer than for individual barchans (Hesse, 2009; Xiao et al., 2023). Such





length and timescales prohibit use of microscopic simulations due to computational costs (Durán et al., 2011), and labora-
tory experiments which cannot reach a sufficient size. Instead, one must turn to an alternative approach, agent-based models
(ABMs) which treat the dunes themselves as the fundamental objects and distil the complexities of the microscopic motion of
sediment into rules for the size-evolution and migration of barchans and interactions between the bedforms.

Several ABMs have been implemented to study barchan swarms (Lima et al., 2002; Durán et al., 2011; Worman et al.,
2013; Génois et al., 2013b) each of which included a different subset of known interaction types. Alongside these, several one-
dimensional ABMs have also been applied to aeolian systems (Parteli and Herrmann, 2003; Lee et al., 2005; Diniega et al.,
2010), however, since barchan swarms are two-dimensional, the results can be very different.

The earliest barchan ABM (Lima et al., 2002) demonstrated that sand flux exchange could lead to spatial structuring while,
since only merging interactions were simulated, the overall size of dunes in the system increased with downwind distance.
Allowing exchange collisions to dominate was shown to produce more realistic global size distributions (Durán et al., 2011)
but did not eliminate the problem that dune size increased with downwind distance (Durán et al., 2011). When fragmenta-
tion collisions, instead, dominate the collision dynamics, clusters form which exhibit a runaway increase in density which is
mitigated only by the the destruction of small dunes and an overall loss of sand that has to be compensated for through the
spontaneous introduction of new dunes throughout the swarm (Génois et al., 2013b). Furthermore, when fragmentation and
sand loss completely dominate the behaviour of a swarm, the largest dunes in the system are the newly injected dunes (Génois
et al., 2013b) which contradicts the intuition that newly formed dunes should be the smallest in a swarm (Elbelrhiti, 2012).
Finally, when collisions result only in merging, while dunes above a certain size are allowed to spontaneously fragment (calve),
the distributions are found to display a large peak at the size at which calving begins to occur and a smaller peak at the size at
which newly calved dunes form with very few barchans at other sizes (Worman et al., 2013). The swarms generated with all
these model versions contrast with real-world swarms which are found to have typical distributions which are similar to a log-
normal (Durán et al., 2009). Thus, none of the existing barchan swarm ABMs has successfully reproduced realistic swarms.
Furthermore, each of the previous works has considered the idealised case of symmetric barchans subject to an idealised uni-
directional wind.

We have developed a new barchan ABM, the Two-Flank Agent-Based Model (TFABM), which has been shown to, not
only, reproduce all of the observed collision behaviour of barchans, but which can also be used to simulate the formation of
asymmetric bedforms under the influence of variable wind regimes (Robson and Baas, 2023). In this paper, we report the
characteristics and behaviours of barchan swarms under unidirectional as well as bimodal winds.



## 2    Model description

The TFABM and its capabilities to replicate all known barchan collision dynamics were introduced in (Robson and Baas, 2023)
and extensive details of the model structure can be found there. However, in this section we review the key features and provide
definitions of the model parameters which are explored in this study.

### 2.1    Dune morphology

The major difference between the TFABM and previous barchan ABMs is the structure of the dunes (agents) themselves. In
the TFABM, each dune is represented by its two flanks, which are able to change size semi-independently of one another. The
position of dunes is recorded as the coordinates of the upwind toe. The central axis of the dune extends in the direction of the
dominant wind which we define to be the positive $x$-direction. The size of the dune is represented by the widths $W_{p,s}$ of the
port and starboard flanks. The widest point of the dune occurs a distance $\lambda_1(W_p + W_s)/2$ downwind of the toe, where we use
$\lambda_1 = 1$ based on comparison with an existing dune dataset (Sherman et al., 2021). The total lengths of the flanks, from toe to tip
of the horn, are assumed to be proportional to their widths $L_{p,s} = \lambda_2 W_{p,s}$ where we use $\lambda_2 = 1.8$ based on real-world dunes
(Sherman et al., 2021). The width of each horn is a linear function of the flank-width $H_{p,s} = \alpha W_{p,s} + \Delta/2$ with $\alpha = 0.05$ and
$\Delta = 4.6\text{m}$ (Hersen et al., 2004; Worman et al., 2013). Finally, the volume of a flank is given by

$$V_{p,s} = \frac{\lambda_2\lambda_3 W_{p,s}^3}{6},\tag{1}$$

where $\lambda_3$ is interpreted as the proportionality coeffecient of flank height/width, we set this to 1/3 such that a symmetric dune
has volume $V_{tot} = 1/40 W_{tot}^3$ in agreement with (Elbelrhiti et al., 2008). For further details of the morphology of dunes in the
TFABM see (Robson and Baas, 2023).

### 2.2    Sand flux

Unlike previous barchan ABMs, the more realistic morphology of dunes in the TFABM allows for variation in the direction
of wind. Incoming sand flux is absorbed by each flank across the windward projection of its width. For oblique winds, this
means that the windward flank will absorb proportionally more of the flux, starving the leeward flank of material. As well as
absorbing material across their wind-facing width, dune flanks also lose material across the width of their horns. Finally, in
asymmetric dunes, there is a transfer of material from the larger to the smaller flank. The rate of this lateral transfer of material
is governed by the model paramater $q_{shift}$ and occurs at a rate proportional to the difference in the streamwise projection of the
flank lengths (note this is not the same as the full length during periods of oblique flow). Combining the absorption, emission,
and lateral shifting flux, the evolution of flank-volume is governed by



$$\frac{dV_{p,s}}{dt} = q_{in}\tilde{W}_{p,s} - q_{sat}H_{p,s} + q_{shift}\sin|90 - \theta|(L_{s,p} - L_{p,s}), \tag{2}$$

where $q_{sat}$ is the saturated sand flux set to $79\text{m}^2\text{year}^{-1}$, the average value of those reported for Tarfaya in (Elbelrhiti et al., 2008), $q_{in}$ is the received influx including both ambient flux and that streaming off of the horns of upwind dunes which overlap the flank, $\theta$ is the wind direction, and $\tilde{W}_{l,r}$ is the flank-width perpendicular to the wind.

### 2.3 Calving

The sizes of the flanks of a dune are coupled only through the third term in equation 2 above. However, when dunes become very asymmetric, this term may become larger than the other terms in equation 2 meaning that spanwise fluxes exceed streamwise fluxes. Since this is no longer the typical behaviour of a barchan, we take this to define a maximum asymmetry threshold for dunes in our model. This threshold occurs when the asymmetry ratio, defined as $\gamma = \max(W_p, W_s)/\min(W_p, W_s)$, exceeds

$$\gamma_{c,shift}(W_p, W_s) = 1 + \frac{\alpha q_{sat}}{\lambda_2 q_{shift} - \alpha q_{sat}} + \frac{\Delta q_{sat}}{2(\lambda_2 q_{shift} - \alpha q_{sat})min(W_p, W_s)}, \tag{3}$$

For highly asymmetric dunes, the typical barchan geometry shown in (Robson and Baas, 2023) can break down if the asymmetry ratio exceeds the threshold

$$\gamma_{c,\lambda} = \frac{2\lambda_2}{\lambda_1} - 1. \tag{4}$$

Thus, combining these two terms, the maximum asymmetry of a dune with flank widths $W_p$ and $W_s$ is

$$\gamma_c(W_p, W_s) = \min(\gamma_{c,\lambda}, \gamma_{c,shift}(W_p, W_s)) \tag{5}$$

When a dune exceeds this maximum asymmetry threshold its two flanks spontaneously break apart and are reformed into symmetric barchans, a process we refer to as calving.

### 2.4 Collisions

One of the reasons that barchans have been studied so extensively is that they migrate rapidly at a rate governed by the the sand-flux conditions and the size of the bedform. We use the commonly accepted expression for migration rate





$$v_{mig} = \frac{cq_{sat}}{W_p + W_s + W_0},  \tag{6}$$

where $c = 45$ (Elbelrhiti et al., 2008) relates to the speed-up of wind passing over the crest of a dune, and $W_0 = 16.6m$ (Elbelrhiti et al., 2008) ensures that the migration rate of small dunes remains finite.

From this expression follows that larger dunes will migrate more slowly than smaller ones, which ensures that collisions are frequent occurrences in barchan swarms. The TFABM uses a simple rule for collisions which is, nonetheless, able to reproduce a wide range of known collision behaviours (Robson and Baas, 2023). Each dune has three associated centres of mass (CoMs), one for each flank and one for the combined dune. The collision algorithm is triggered whenever one of these centres of mass intersects with the footprint of another dune. When that condition is met, the following algorithm is used to determine the output of a collision:

1. Intersecting flanks merge together

2. The volume of merged flanks is used to calculate an effective flank width $W_{merged}$

3. A non-intersecting flank with width $W_i$ will also join the merged flanks if $W_{merged}/W_i \leq \gamma_c(W_{merged}, W_i)$

4. If $W_{merged}/W_i > \gamma_c(W_{merged}, W_i)$ the non-intersecting flank forms into a separate symmetric barchan in the same manner as during calving.

5. Finally, the flanks which merged together form a barchan with preserved centre of mass and an asymmetry ratio determined by the relative mass that was initially to the left and right of the centre of mass.

The algorithm itself can produce between 1 and 3 output dunes although calving may occur on the following timestep such that a collision may effectively produce up to 4 outputs. For more information on the phase space of the TFABM collision rule see (Robson and Baas, 2023).

## 2.5 Boundary conditions and dune injection

The simulation space has a downwind length of 8km and a central 5km-wide strip for accommodating the swarm, flanked on either side with additional 5km widths of domain to allow for lateral movements of dunes under oblique winds. At the upwind boundary over the entire width (15km) of the simulation space, we supply a free flux $q_0 = 0.25q_{sat}$ (Elbelrhiti et al., 2008). This flux is projected downwind until it is intercepted by a dune, downwind of which the free flux is zero.

In this work, we do not use periodic boundaries, thus, once dunes exit the simulation space they are lost. To account for this, new dunes enter the simulation space at the upwind boundary. The rate of injection of new dunes is governed by assuming





that just upwind of the domain, there is a number density $\rho_0$ of symmetric barchans each with a width $W_{eq} = \Delta/(q_0 - \alpha q_{sat})$,

which corresponds to the known unstable equilibrium size (Hersen et al., 2004; Worman et al., 2013). Dunes are injected only into the central 5km wide strip of the simulation space to ensure that dunes do not reach the lateral borders. The length of a timestep was set to $\delta_t = 1/12$ years $\approx 30$days. As we will see, the smallest dunes in the system are typically the newly created bedforms with width $W_{eq}$. With our choice of $q_{sat}$, $c$, and $W_0$, (see equation 6) the turnover time (time taken for a dune to migrate its own length) of barchans with width $W_{eq}$ is $t_{turnover}(W_{eq}) = 84$days, meaning that our choice of timestep is still

much shorter than the shortest timescale relevant to the dunes in our simulations. The number of dunes entering the system each timestep is then determined by

$$N_{enter} = \frac{W_{simulation}}{3}\rho_0 v_{mig}(W_{eq})\delta_t. \tag{7}$$

### 2.6 Unimodal Wind

For the first series of simulations, we use a Gaussian distribution for the direction of the wind centred around $0°$. We experimented with different values of the standard deviation but found that, provided the standard deviation remains relatively small (we tested up to $5°$), the exact value is not important. Therefore, all of the simulations shown in this section were performed using a standard deviation of $3°$. Since the wind angle is normally distributed, this standard deviation ensures that 99.7% of the time, the angle of the wind is within $\pm 9°$ of the primary mode. Under this wind regime we investigated the effect of changes

in $q_{shift}$ and $\rho_0$ as well as the effect of a sudden change in a boundary condition on a stable swarm.

### 2.7 Varying $q_{shift}$

As shown in (Robson and Baas, 2023) the rate of equilibration between the flanks, represented by the flank balancing flux, $q_{shift}$, has a major impact on the output of collisions. This is because a higher value of $q_{shift}$ means a lower maximum

asymmetry ratio (see equation 3), thus increasing $q_{shift}$ leads to a larger portion of the collision phase space resulting in fragmentation collisions (Robson and Baas, 2023). However, the properties of the entire phase space of collisions may not be relevant in a real-world swarm since it has been shown that barchans in swarms show spatial structuring, such as preferential alignment (Elbelrhiti et al., 2008; Lima et al., 2002), which may result in collisions occurring more frequently in a particular part of the phase-space. Therefore, to investigate the effects of varying $q_{shift}$, we fixed $\rho_0 = 37$ km$^{-2}$ and performed simu-

lations with $q_{shift} = 0, 0.1q_{sat}, 0.2q_{sat}, 0.3q_{sat}$, and $0.4q_{sat}$. Each of these simulations was performed over a timescale long enough for the entire 8km length of the simulation space to be filled and the properties of the dunes to have stabilised, this meant simulations of $>300$ years in all cases. In figure 1 we show the final states of these simulations alongside plots showing that the population size and mean width had stabilised during the simulations. Animations showing the entire evolution of the swarms can be found in Robson (2023).








**Figure 1.** The final states of the simulated swarms and plots showing that the mean dune width and population size had stabilised for $\rho_0 = 37$ km$^{-2}$ and $q_{shift}/q_{sat} =$ a) 0.0, b) 0.1, c) 0.2, d) 0.3, and e) 0.4.



For lower values of $q_{shift}$ we observe a smaller total number of dunes in the system, and therefore a lower overall density, while the average size of dunes in simulations with lower $q_{shift}$ is greater than when $q_{shift}$ is high. To explain this, we show in figure 2 how changing $q_{shift}$ affects the properties of collisions. We observe that, as $q_{shift}$ increases, the collision behaviour switches from exchange/merging dominated to fragmentation dominated with the transition occurring between $q_{shift}/q_{sat} =$

0.1 and 0.2. For $q_{shift} \geq 0.2q_{sat}$ the relative frequencies of the different collision types begin to stabilise which coincides with the the average lateral offsets and collision width ratio for which the collision types occur also reaching stable values. This convergence to a particular type of collision dynamics at $q_{shift} \gtrsim 0.2q_{sat}$ was likely the reason that the population sizes of swarms were approximately equal above this threshold (see figures 1c-e). We also show in figure 2 that the average position of collisions fell in the upwind half of the swarm ($< 4km$) suggesting that there might be an inhomogeneity in the position of the

dunes in the streamwise dimension. We observed no inhomogeneity in the spanwise coordinate of collisions in unidirectional simulations.





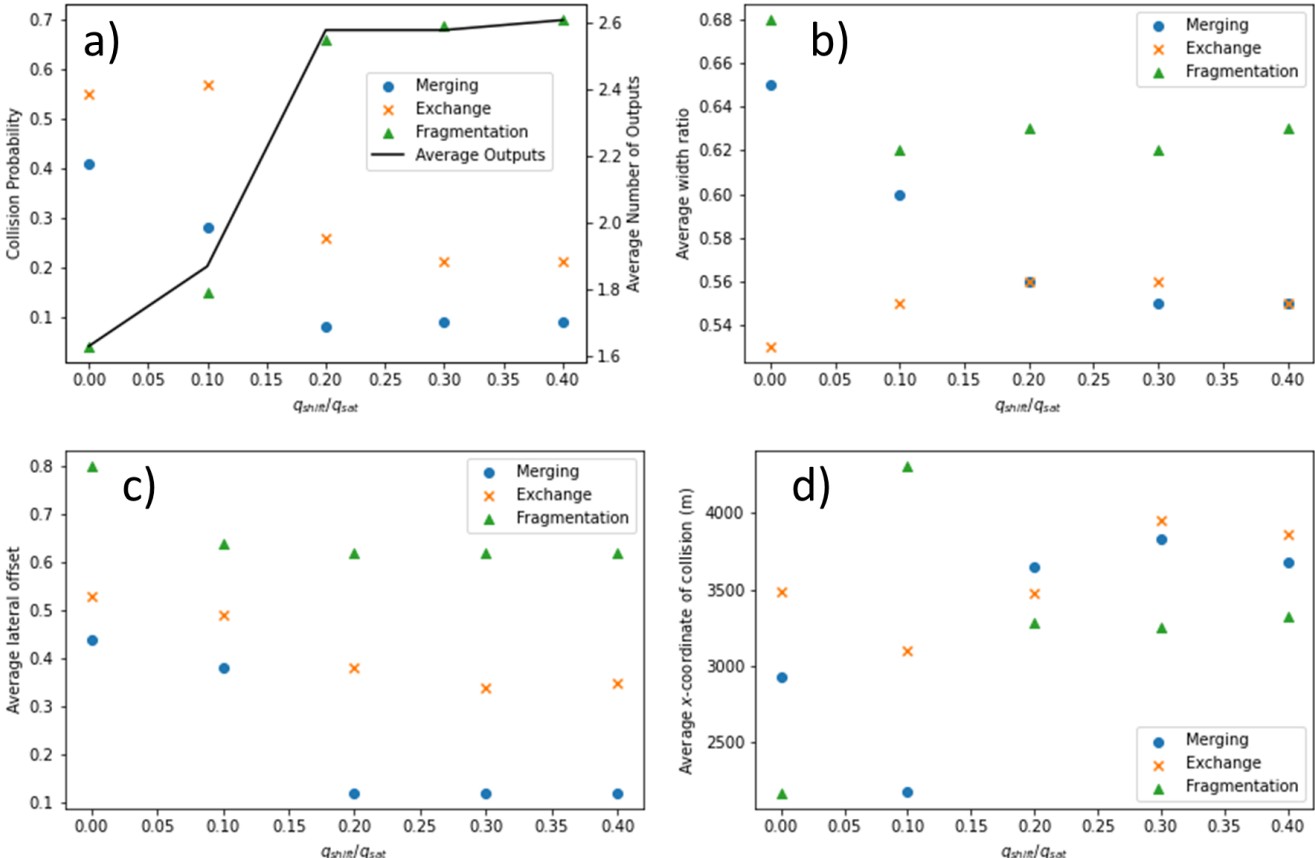

**Figure 2.** Properties of collisions observed in simulations with $\rho_0 = 37$ km$^{-2}$ and $q_{shift}/q_{sat} = 0.0, 0.1, 0.2, 0.3,$ and 0.4. a) The relative frequencies at which the different collision types occurred and the expected number of outputs of a collisions. b) The average width ratio of colliding dunes for each collision type. c) The average lateral offset of colliding dunes normalised by their mean width for each of the collision types. d) The average $x$-coordinate (streamwise) at which collisions of different types occurred.

It has been found that the sizes of dunes in real-world swarms remain stable in the longitudinal dimension (Elbelrhiti et al., 2008; Durán et al., 2011), however, this is not something that previous agent-based models have been able to reproduce (Lima et al., 2002; Durán et al., 2011; Worman et al., 2013). In figure 3 we show that, in contrast, we are able to produce stabilised sizes of dunes when $q_{shift} > 0.1q_{sat}$ with the swarms exhibiting a trend whereby the sizes of dunes increases until around 2-3km after which the size distribution remains approximately steady. For $q_{shift}/q_{sat} \leq 0.1$ the stabilisation of sizes did not occur within the simulation space, though it is possible that it may have done further downwind. Such an increase in size has been observed in narrow real-world swarms but not large swarms as simulated here (Elbelrhiti et al., 2008; Durán et al., 2011).

As seen in figure 3, the dune number density displays a more complex behaviour. The density initially decreases sharply, reaching a local minimum at around 1km. Continuing downwind from this minimum, all five swarms show that density then



increases, reaching a local maximum between 2-3km. Finally, the density then steadily decreases with downwind distance. The final steady decrease in density coincides approximately with the stabilisation of dune size. The generally decreasing trend of density with downwind distance explains why the average longitudinal dimension of collisions is typically $< 4$km as the greatest number of collisions will occur in the upwind portion of the swarm where the density it highest.

As well as having a significant impact on the collision dynamics, we observe that varying $q_{shift}$ dramatically changes the distribution of dune asymmetry and the rate of calving in the simulated swarms as shown in figure 4. As we increase the, $q_{shift}$ the width of the asymmetry distribution decreases as can be seen from the decrease in the standard deviation of the distributions. Populations on barchans in Tarfaya, Mauritania, and Mars were previously measured (Robson et al., 2022) and found to have asymmetry ratio standard deviations in the range 0.089-0.149, which matches closely with our simulations of $q_{shift}/q_{sat} = 0.1$ and 0.2 which saw respectively $\sigma = 0.14$, 0.09.

While the asymmetry of dunes follows a clear trend with increasing $q_{shift}$, the dependence of calving rate on $q_{shift}$ is non-monotonic. For low $q_{shift}$ the rate of calving is high as there is no material exchange between the flanks which can prevent runaway asymmetry growth, as exhibited by the wide asymmetry distributions for the lower values of $q_{shift}$. This means that many dunes calve due to asymmetry threshold imposed by the constraints of the geometry of the dunes. On the other hand, when $q_{shift}$ is high, any asymmetry induced in the dunes through oblique winds, collisions, asymmetric fluxes from neighbouring dunes, will lead to calving as $\gamma_{c,shift}$ (see equation 3) decreases rapidly with increasing $q_{shift}$. Therefore for both high and low $q_{shift}$ the rate of calving is high, whereas intermediate values of $q_{shift}$ display the lowest rates of calving.





**Figure 3.** The mean width and dune density in 500m cross-sections averaged from measurements at the end of each year once the swarm properties stabilised $q_{shift}/q_{sat} = 0$, 0.1, 0.2, 0.3, and 0.4 in a)-e) respectively. The grey areas represent one standard deviation.







**Figure 4.** Histograms of dune asymmetries defines at the ratio of port and starboard flank widths are shown in a)-e) for respectively $q_{shift}/q_{sat} = 0,$, 0.1, 0.2, 0.3, and 0.4. f) Shows how standard deviation of dune asymmetries and the number of calving events per year per dune varied with $q_{shift}$

Finally, dunes have been observed to typically align with the horns of their upwind neighbours (Lima et al., 2002; Bagnold, 1941) which leads to a pair of peaks in the distribution of the lateral offset of nearest downwind neighbours when normalised by the width of the upwind dune (Elbelrhiti et al., 2008). In figure 5 we show the distributions of this normalised lateral offset of nearest downwind neighbours in our simulations. For comparison, examples from six real-world swarms (Robson et al., 220 2022) are shown in figure 6. Both the real-world and simulated data show that dunes typically do not align with the centre





of their upwind neighbour (a lateral offset of zero) and exhibit strong peaks, which for the simulated swarms and terrestrial real-world swarm (figures 6 a)-d)) occur at offsets $\sim \pm 1$ (note that +1 indicates that the downwind neighbour is aligned with the starboard flank of the upwind dune and -1 indicates that it is aligned with the port flank). Unlike the real-world swarms however, we observe a secondary peak at slightly larger offsets and we do not observe a significant difference between the cases when the upwind dune is larger than its downwind neighbour ($W > W_{downwind}$) versus when the downwind neighbour is the larger bedform ($W \leq W_{downwind}$).








**Figure 5.** The lateral offset of the nearest downwind neighbour of dunes normalised by the width of the upwind dune for $q_{shift}/q_{sat} = 0$, 0.1, 0.2, 0.3, and 0.4 in a)-e) respectively. The figures on the left shown the distributions each year with the most recent years shown in the darker colours. On the right, are the time averaged distributions for the two cases when the upwind dune is smaller or larger than the downwind neighbour.





**Figure 6.** The normalised lateral offset of the nearest downwind neighbours in the populations a) Tarfaya 1 b) Tarfaya 2, c) Tarfaya 3, d) Mauritania, e) Mars 1, and f) Mars 2 which were described in Robson et al. *Physica A* (2022).



## 2.8 Varying $\rho_0$

The previous section shows that varying $q_{shift}$ leads to significant differences in the simulated swarms through changing the collision, asymmetry, and calving behaviour of the dunes. However, the rate of injection of new dunes at the upwind domain boundary was kept at $\rho_0 = 37$ km$^{-2}$ in those simulations. Real-world swarms however, are observed to have a wide range of densities (Elbelrhiti et al., 2008). Figure 7 shows results from simulations where we set $q_{shift} = 0.2 q_{sat}$ and used $\rho_0 = 12, 24,$ 37, 61km$^{-2}$ (these correspond to 5%, 10%, 15% and 25% of the density of Tarfaya 1 in (Robson et al., 2022)), revealing that

as injection density increases, so too does the overall number of dunes in the swarm, while, as in the case of increasing $q_{shift}$, the mean sizes of dunes decreases. While these densities are lower than some swarms (Robson et al., 2022), there are many examples of swarms with similar densities to these (Elbelrhiti et al., 2008).





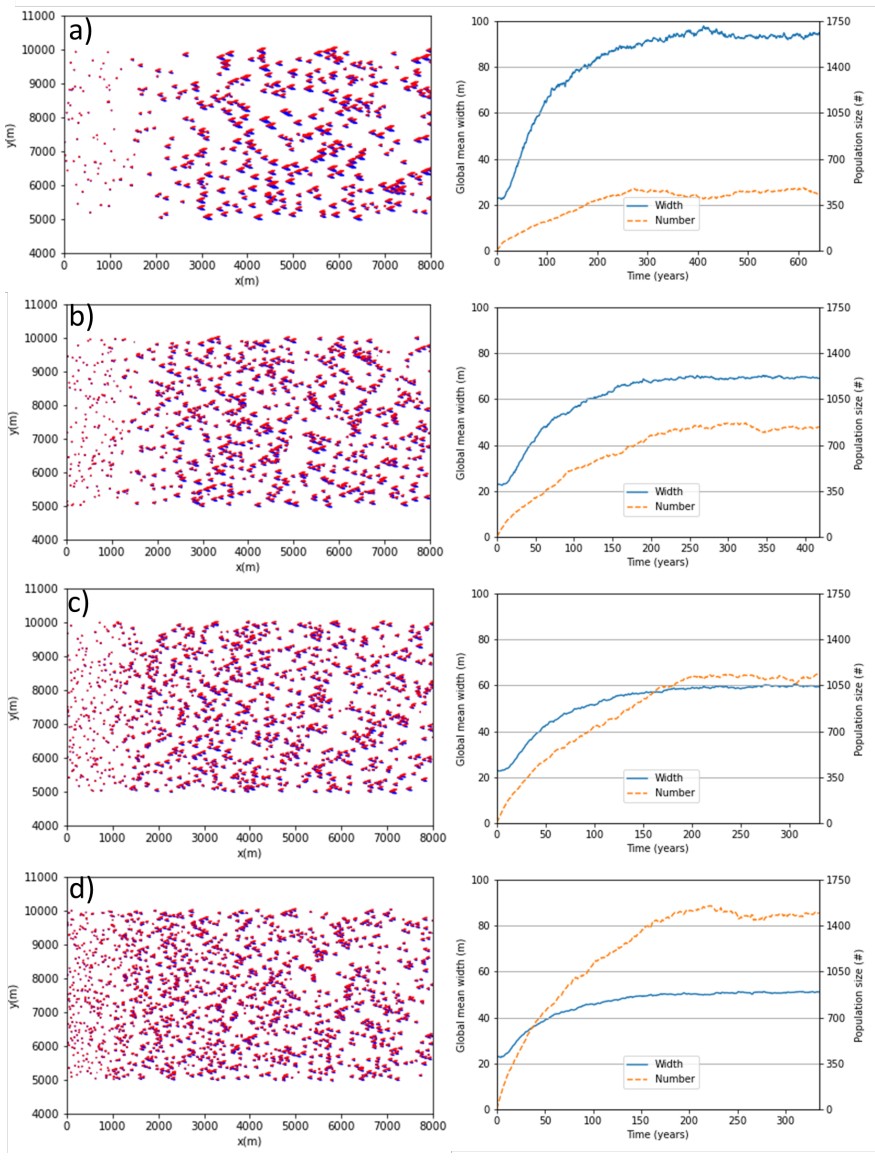

**Figure 7.** The final states of simulations and stabilisation of the dune number and mean size with time for swarms with $q_{shift} = 0.2 q_{sat}$ and $\rho_0 = 12, 24, 37, 61 \mathrm{km}^{-2}$ in a)-d) respectively.

In the previous section, we showed that the changes in the final states of the swarms could be explained because by changing
$q_{shift}$ we had significantly altered the collision dynamics. However, we observe that the properties of collisions in the constant $q_{shift}$ simulations remain the same, except for the average streamwise position at which the collisions occur, as shown in figure 8a). Stability of the collision behaviour is to be expected since it has been shown that the TFABM collision rule is primarily controlled by $q_{shift}$ (Robson and Baas, 2023). On the other hand, we see that increasing $\rho_0$ has a significant impact on the $x-$coordinate of collisions. Collisions are observe to occur most frequently in the downwind portion of the swarm (indicated



by an average position >4km) for low values of $\rho_0$ but as the injection density increases the position of collisions shifts significantly further upwind. This suggest that the injection density has a strong impact on the longitudinal density profile of the swarms. Since $qshift$ remained constant in these simulations, the way in which the asymmetry ratio could vary would be if that parameter is also affected by the overall density of dunes. However, we see only a very minor difference in the width of the asymmetry distribution as shown in figure 8b) alongside the rate of calving. We do observe a slight increase in the rate

of calving, however this is likely only a result of the increased rate of collisions which can directly lead to calving events occurring (Robson and Baas, 2023).

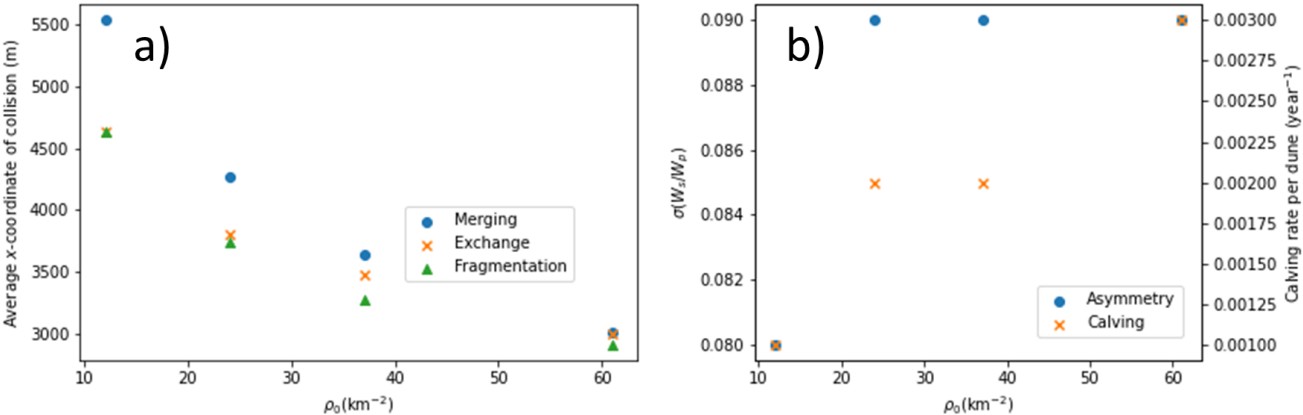

**Figure 8.** The variation of a) the streamwise coordinates of collisions and b) the standard deviation of the bilateral asymmetry distribution and rate of calving in swarm simulations with $q_{shift} = 0.2q_{sat}$ and $\rho_0 = 12, 24, 37, 61\text{km}^{-2}$.

The longitudinal variation of dune size and number density are shown in figure 9. In all cases, we observe that the sizes of dunes in the system stabilise with downwind distance and for $\rho_0 \geq 24\text{km}^{-2}$ we see a similar pattern where the density

decreases to a minimum before increasing slightly and then decreasing steadily. For the sparsest swarm, we observe the same initial trend but do not see the steady decrease with density remaining approximately stable or slightly increasing from 4-8km downwind, which explains why the average $x-$coordinate of collisions in the $\rho_0 = 12\text{km}^{-2}$ was >4km, since the downwind portion of that swarm has the highest density of dunes and therefore will have the highest rate of collisions. The portion of the density profile between 0 and $\sim$3km shows the greatest variation in density, this might be interpreted as a changing boundary

condition which has been suggested should result in a greater rate of dune interactions (Marvin et al., 2023). However, we do not observe a significant rate of collisions in that part of the profile but rather observe that collisions generally occur wherever density is highest. We explore changing boundary conditions later on in this section.





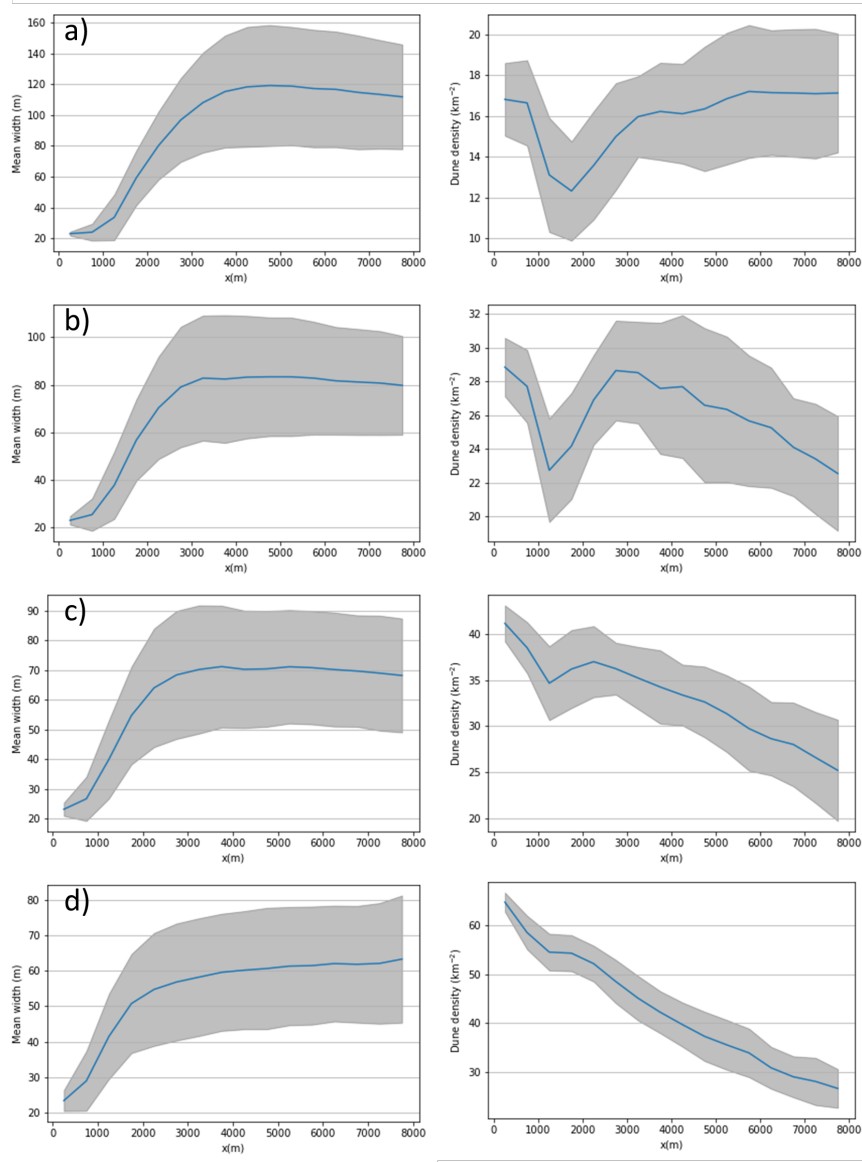

**Figure 9.** The mean width and dune density in 500m cross-sections averaged from measurements at the end of each year once the swarm properties stabilised. The grey areas represent one standard deviation. For simulations with $q_{shift}/q_{sat} = 0.2$ and $\rho_0 = 12, 24, 37, 61\text{km}^{-2}$ in a)-d) respectively,

Finally, having performed simulations varying either $q_{shift}$ or $\rho_0$ we can begin to analyse which of these parameters has the
most significant impact on the observed swarms. In figure 10 we show how the size and number density of dunes is affected by varying each of these parameters. The results show that, while there are visible gradients in each dimension, the more significant changes to the system occurr when varying $\rho_0$.



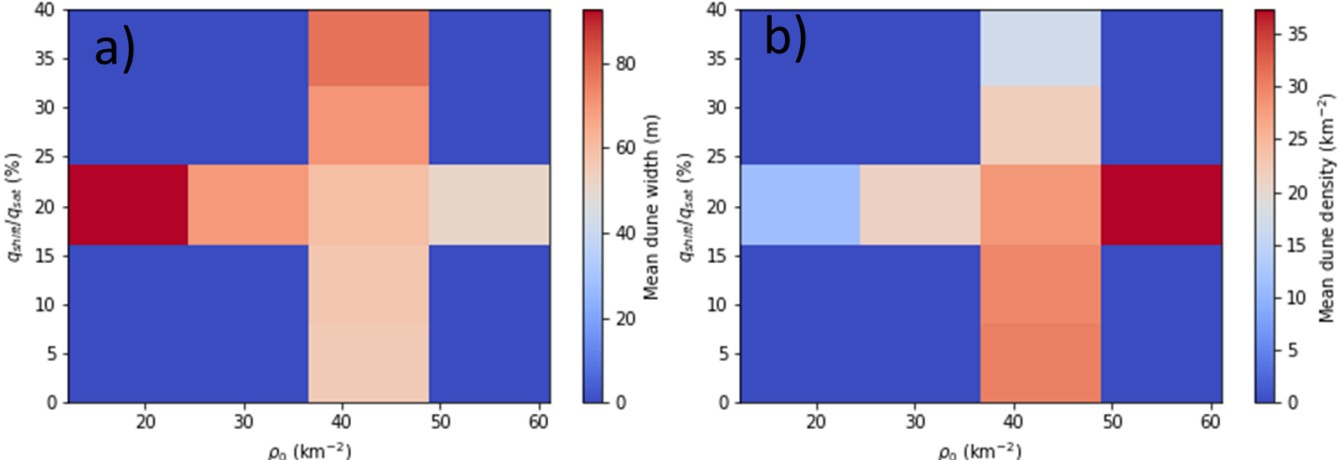

**Figure 10.** The variation of a) mean dune size and b) mean dune number density with $q_{shift}$ and $\rho_0$ for unimodal swarm simulations. The properties of interest in each figure are indicated by the colourbar, with values of zero indicating areas of the phase-space which were not explored in the simulations.

### 2.9 Changing boundary conditions

Barchan swarm dynamics might reflect sudden changes in boundary conditions such as an increase or decrease in upwind sediment supply. We carried out simulations which used as their starting point the final state of the $q_{shift}/q_{sat} = 0.2$, $\rho_0 = 37\text{km}^{-2}$. We then changed the injection density, in one case to $12\text{km}^{-2}$ and in another to $61\text{km}^{-2}$ and resumed the simulations. In figure 11 we show the simulated swarm and the distribution of widths in cross-sectional intervals at several time steps after $\rho_0$ was decreased to $12\text{km}^{-2}$. The width distribution shows how an adjustment to a higher mean dune width is migrating longitudinally through the swarm until the whole swarm reaches a new steady state that is similar to that shown in figure 9a). The equivalent plot for the case where $\rho_0$ was increased to $61\text{km}^{-2}$ is shown in figure 12 where this time we see that the mean dune width becomes smaller until reaching a similar distribution to that shown in figure 9d).



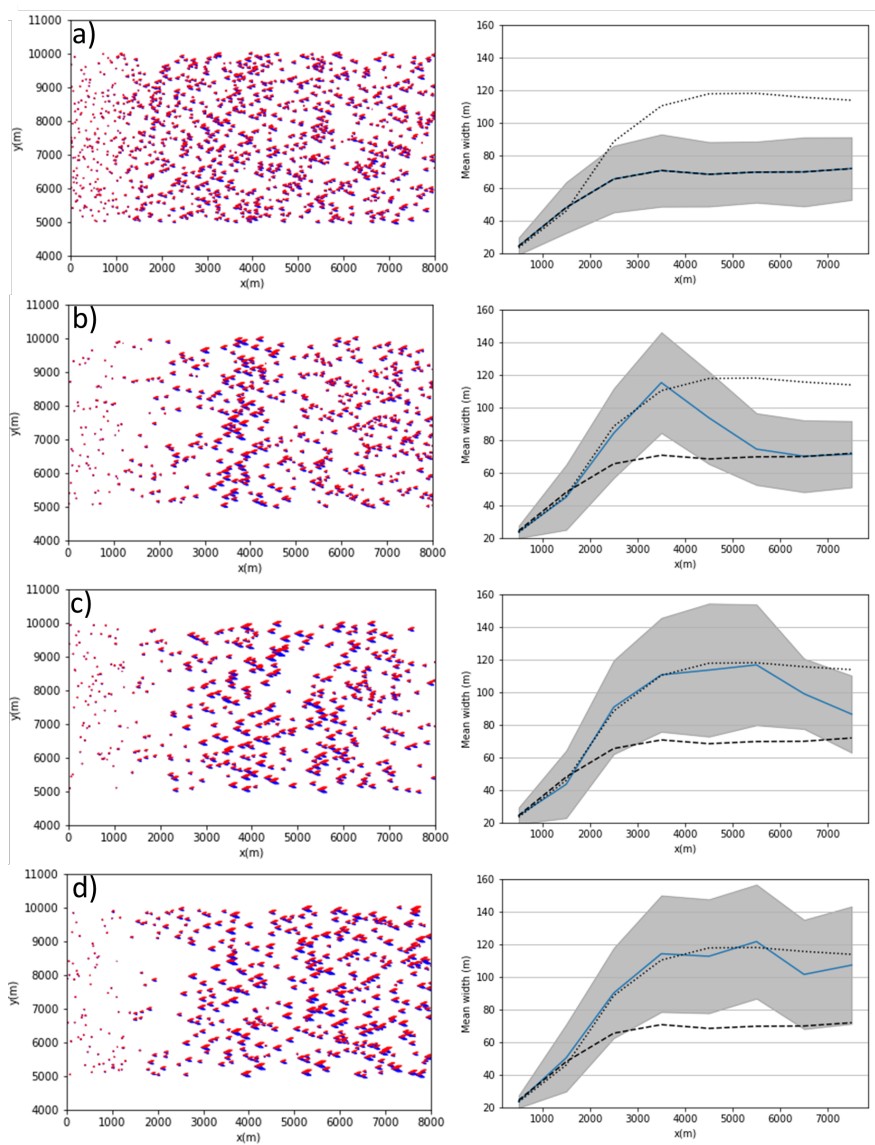

**Figure 11.** Snapshots and mean widths of dunes in 1km intervals of a swarm which stabilised with $q_{shift}/q_{sat} = 0.2$, $\rho_0 = 37\text{km}^{-2}$ a) initial steady state b) 100 year, c) 200 years, d) 400 years after $\rho_0$ was changed to $12\text{km}^{-2}$. The grey areas mark the standard deviation of the width. The dashed line indicates the initial profile before the boundary condition was changed and the dotted line indicates the long-time average from swarm simulations with $\rho_0 = 12\text{km}^{-2}$.





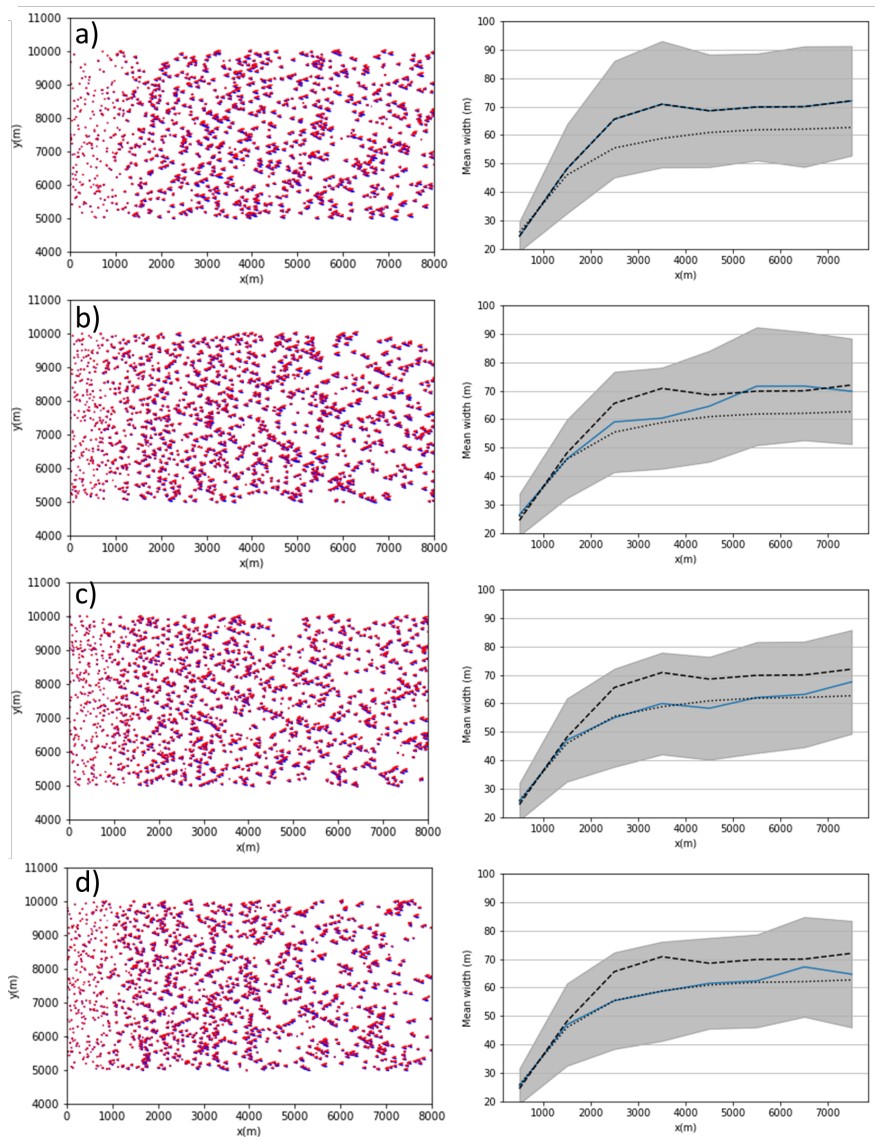

**Figure 12.** Snapshots and mean widths of dunes in 1km intervals of a swarm which stabilised with $q_{shift}/q_{sat} = 0.2$, $\rho_0 = 37\text{km}^{-2}$ a) initial steady state b) 100 year, c) 200 years, d) 400 years after $\rho_0$ was changed to $61\text{km}^{-2}$. The grey areas mark the standard deviation of the width. The dashed line indicates the initial profile before the boundary condition was changed and the dotted line indicates the long-time average from swarm simulations with $\rho_0 = 12\text{km}^{-2}$.

The evolution of the global mean width and population size for the entire duration of the simulations, including the 330 years taken to stabilise with the initial conditions, are shown in figure 13. The plots reveal the steady increase and stabilisation 280 of the swarm under the initial conditions and then a sudden change in the properties when the boundary condition changes at T=330 years as the mean size and dune number adjust, eventually stabilising after a further 200-300 years. The trend is much



clearer in the case where we reduced the injection density, perhaps because the difference between the initially stable state and the final stable state is much greater than for the case of increasing injection density. We also show in figure 13 that the average number of collisions adjusts to the change in boundary condition via a gradual relaxation to a new steady state, whereas we

see no sharp spike in collision rate during the adjustment period, contrary to what has recently been suggested by Marvin et al. (Marvin et al., 2023).

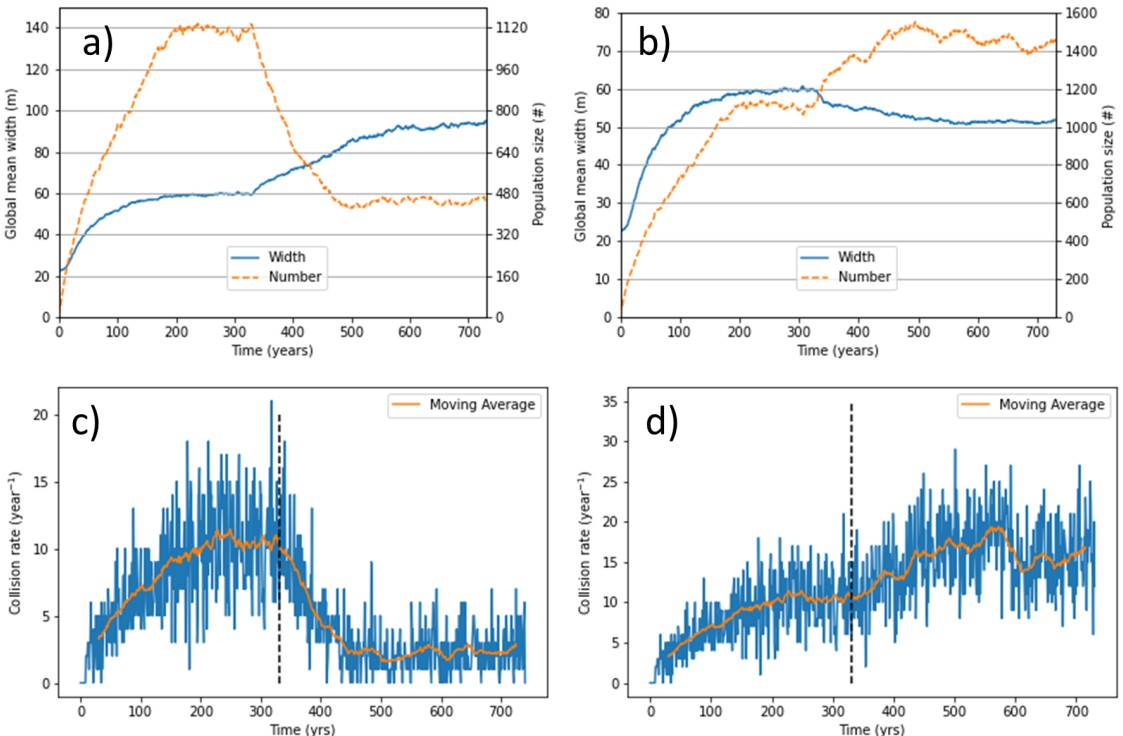

**Figure 13.** The mean sizes and population sizes as a function of time for simulations where $q_{shift}/q_{sat} = 0.2$ and $\rho_0 = 37\text{km}^{-2}$ for 330 years before changing to a) $\rho_0 = 12\text{km}^{-2}$ b) $\rho_0 = 61\text{km}^{-2}$. c) and d) show the average number of collisions per dune with the orange line showing a 30 year moving average.

### 2.9.1 Bimodal Simulations

While the simulations in the previous section all simulated a unimodal wind, many real-world swarms are exposed to a sec-

ondary mode for some parts of the year, something that we can simulate using the TFABM. We performed simulations with $\rho_0 = 37\text{km}^{-2}$ and $q_{shift} = 0.2q_{sat}$ where for three quarters of the year, the wind follows the dominant mode, set to a Gaussian distribution centred on $0°$ with standard deviation of $3°$, while for the remaining season, the wind direction was set to a Gaussian distribution with the same standard deviation but centred on an angle $\theta_b$ for which we used $36°$, $72°$, $108°$, and $120°$.



We were unable to generate stable swarms when using angular separation of $> 120°$ between the primary and secondary modes.


In figure 14 we show the final states of simulations for the different modal angular separations with $\rho_0 = 37\text{km}^{-2}$ and $q_{shift} = 0.2q_{sat}$ alongside plots indicating that both the mean size and dune number stabilise during the course of the run. The swarms that are subject to modes with an acute angular separation appear broadly similar to the unimodal simulations, although the dunes have migrated outside of the central 5km-wide strip in which they remained approximately confined during

the unimodal runs. As the angular separation of the modes increases, we see that the bottommost edge of the swarm develops into a continuous stream of dunes stretching from the upwind boundary to the downwind. In the obtuse angle swarms, the dunes which are not close to this edge are small in number and very large. The obtuse angle runs also had to be conducted for much longer time periods ($> 1500 years$) due to the presence of a small number of very large dunes spanning the width of the swarm which formed at the start of the run, before the stream of dunes had had time to form. Since these dunes were large, they

took a very long time to travel the 8km and exit the simulation space and hence, we had to allow the simulations to progress for much longer than in other runs.



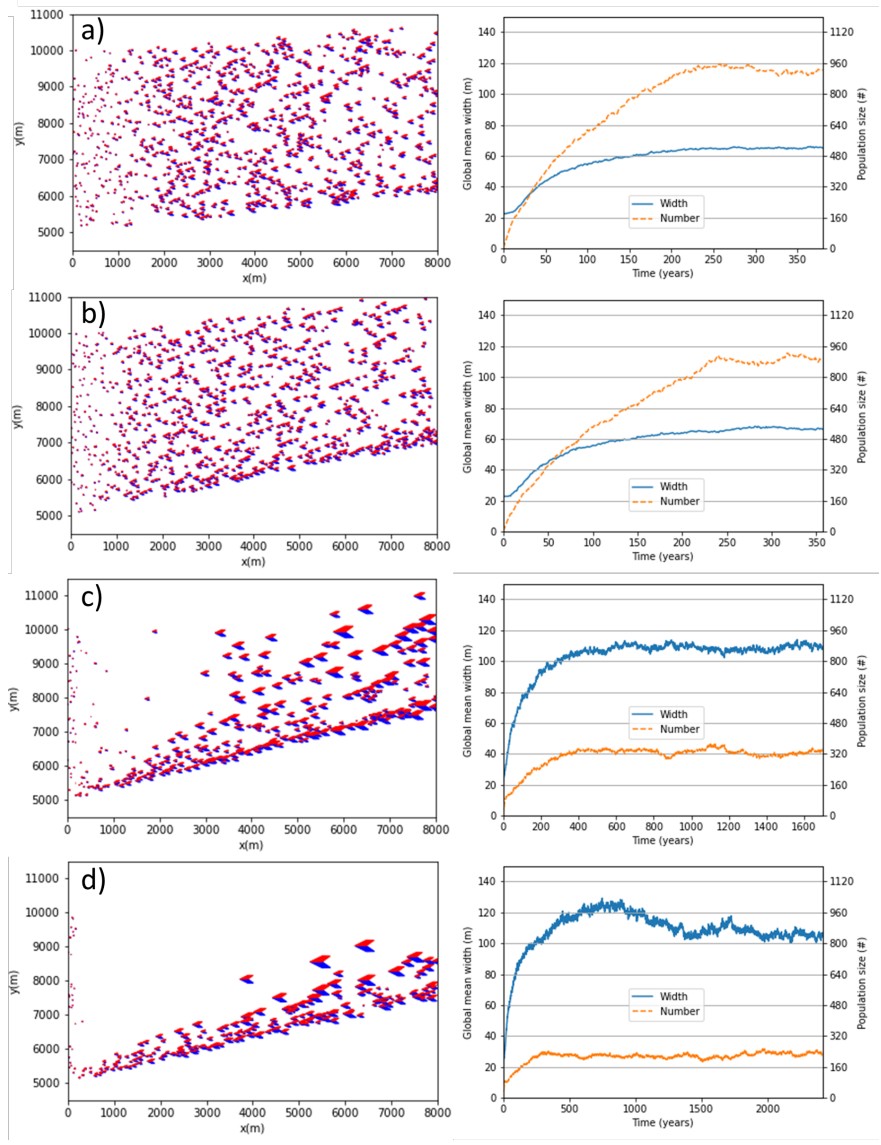

**Figure 14.** Final states and stabilisation of mean size and dune number for $\rho_0 = 37 \mathrm{km}^{-2}$ and $q_{shift} = 0.2 q_{sat}$ and secondary wind mode for the last quarter of each year with angular separation a) $36°$, b) $72°$ c) $108°$, and d) $120°$.

As shown in figure 15, the longitudinal size and density profiles of the swarm under acute angular separation regimes display very similar characteristics to the unimodal simulations. On the other hand, the density profiles for the swarms under obtuse

modal separation give very different profiles, eventually with the rate of decrease of density slowing with increasing longitudinal distance. The mean size in the obtuse swarms appears to be plateauing, as for the acute separation swarms, but has not yet reached a steady value.





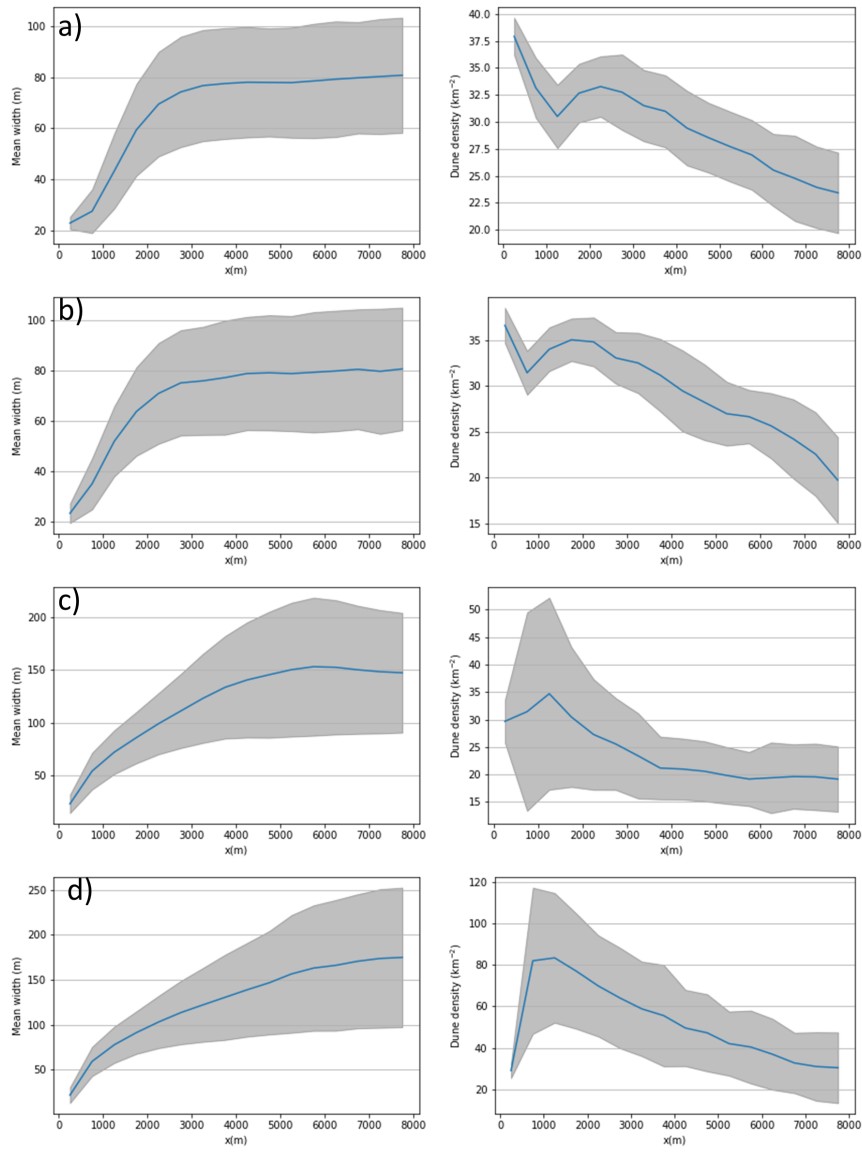

**Figure 15.** The mean width and dune density in 500m transects averaged from measurements at the end of each year once the swarm properties stabilised. The grey areas represent one standard deviation. For simulations with $q_{shift}/q_{sat} = 0.2$ and $\rho_0 = 37\text{km}^{-2}$ and angular separation a) $36°$, b) $72°$ c) $108°$, and d) $120°$.

Although $q_{shift}$ is the major control on collision outcomes in the TFABM, it has also been shown that the relative asymmetry of colliding dunes (Robson and Baas, 2023) and direction of migration (Bo and Zheng, 2013) can also control barchan interactions. In figure 16 we show that changing to a bimodal wind regime has affected the relative probabilities of collisions with merging becoming more likely and fragmentation less frequent (though still dominant) leading to an overall reduction in the average number of outputs of a collision as the angular separation of the modes increases. The lateral offset of collisions





were not seen to vary greatly, therefore the major change in the relative frequencies of the collision types is as a result of
changes to the average width ratio with merging being able to take place for dunes which are more similar in total size as seen
in figure 16b).

As in the case of unimodal collisions, the swarms with fewer dunes have a lower rate of collisions since the overall density
is lower. Finally, we see now that the average position of collisions has been displaced from the centre of the swarm in both
the streamwise (figure 16c)) and spanwise (figure 16d)) dimension. For acute angular separation, we see that the spanwise
coordinate shifts to higher values, since the dunes have migrated in that direction. For obtuse angular separation, however,
the average position of collisions shifts in the other direction, since most of the dunes are confined to a narrow stream on the
starboard edge of the swarm.

In the streamwise dimension, with acute angular separation of modes, collisions occur more frequently in the upwind region
where the density is higher with all three types of collision occurring in similar locations. However, when the separation is
obtuse, the average position of exchange and fragmentation collisions shifts further downwind while merging shifts in the
opposite direction. A likely reason for this is that, in the upwind portions of the swarms subject to the obtuse winds, the dunes
are closely aligned with one another, since they are confined in the narrow stream on the starboard edge of the swarm. Aligned
collisions typically result in merging (Robson and Baas, 2023), meaning that merging occurs in the upwind regions but ex-
change and fragmentation can only occur further downwind where the dunes begin to spread apart laterally further downwind.





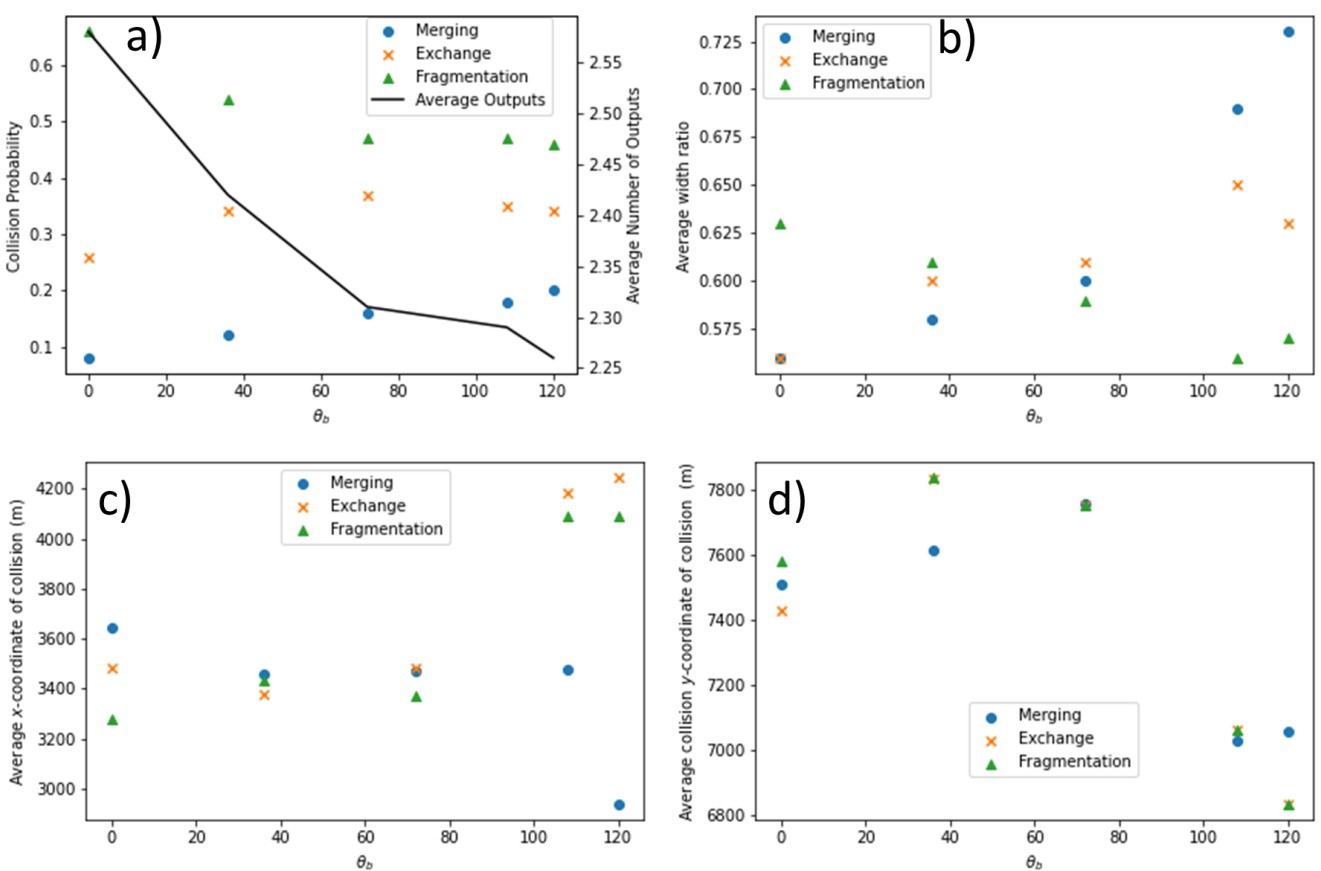

**Figure 16.** The variation of the properties of collision in swarm simulations with $q_{shift} = 0.2q_{sat}$ and $\rho_0 = 37 \mathrm{km}^{-2}$. a) The relative frequencies at which the different collision types occurred and the expected number of outputs of a collisions. b) The average ratio of widths of colliding dunes for each collision type. c) The average $x$-coordinate (streamwise) at which collisions of different types occurred. d) The average spanwise coordinate ($y$) at which the different types of collisions occurred.

It was shown in (Robson and Baas, 2023) that, using the Two-Flank Agent-Based model, under a bimodal wind where the secondary mode has acute angular separation from the dominant mode asymmetry grows according to the Bagnold model (Bagnold, 1941; Bourke, 2010) whereas for obtuse angular separation it grows according to the Tsoar model (Tsoar, 1984; Tsoar and Parteli, 2016; Bourke, 2010). Similar findings have also been reported using other types of modelling (Parteli et al., 2014b; Lv et al., 2016).

The distributions of asymmetry are shown in figure 17. One would expect that under the Bagnold model for asymmetry growth (Bagnold, 1941) the starboard flank would grow predominantly under our bimodal winds leading to $W_p/W_s < 1$ while under the Tsoar model we would see $W_p/W_s > 1$. In figure 17f), however, we see that in all of the bimodal runs the





average of $W_p/W_s$ is less than one, indicating that overall the asymmetry of dunes is growing predominantly according to the Bagnold-model (Bagnold, 1941; Bourke, 2010). We do observe, however, that the mean asymmetry varies with the angle of the secondary mode. Specifically, for acute separation, as the angle increases so too does the deviation of the mean from one. Once

the angular separation becomes obtuse, however, the asymmetry becomes less pronounced with increasing angle. This suggest that the growth of asymmetry is most pronounced when the secondary and primary modes are roughly perpendicular. Although this hints that the relationship may be sinusoidal, we can see that this is not the case by comparing the average asymmetry for $\theta_b = 72°$ and $108°$ which show that the dunes under the obtuse wind are less asymmetric than those under an acute separation. This perhaps suggests that Tsoar-like asymmetry growth (Tsoar, 1984; Tsoar and Parteli, 2016) does occur in some of the

dunes when the separation is obtuse. This possibility is discussed in more detail in the Discussion.

We also show in figure 17 that the rate of calving varies with the angle of the secondary mode, with the greatest rate of calving observed when $\theta_b = 72°$ which is also when the dunes are most asymmetric. We also show that the dunes become more asymmetric and calve more frequently in the case of $q_{shift} = 0.1q_{sat}$ when we change from a unimodal simulation to a bimodal one with angular separation $36°$.





**Figure 17.** Histograms of dune asymmetries defines at the ratio of port and starboard flank widths are shown in a) for $\theta_b = 36°$ with $q_{sat}/q_{shift} = 0.1$ and then b)-e) for $q_{sat}/q_{shift} = 0.2$, and respectively, $\theta_b = 36°$, $72°$, $108°$, and $120°$. f) Shows how the average dune asymmetry and the number of calving events per year per dune varies with $\theta_b$ ($\theta_b = 0°$ indicates unimodal simulations) for $q_{sat}/q_{shift} = 0.2$ and 0.1.



It was shown in previous mean-field modelling that the relative frequencies of collisions and calving can have significant impacts on the behaviour of a swarm (Robson et al., 2022). We have seen that changing $q_{shift}$, $\rho_0$, and $\theta_b$ all have an impact on the rate of calving, but it is important to also examine how these parameters affect the rate of collisions. In figure 18a) we show that the rate of collisions can be predicted entirely from the density, $\rho$, of the swarm irrespective of whether the simulation was unimodal or bimodal, with all of the simulations performed here falling onto a single curve $\propto \rho^2$. On the other hand, no such

trend is seen for the rate of calving with each of the parameters we varied having an impact on its rate. We find that $q_{shift}$ has the most noticeable effect on the calving rate as shown in figure 18b), however there is still a great deal of variation caused by the other parameters. Finally, in figure 18c) we show the relative frequencies of collisions and calving with $q_{shift}$. It is important to mention here, however, that since collisions can trigger a calving event (Robson et al., 2022), the rate of calving may be somewhat inflated in this figure.

As well as affecting collisions, asymmetry, and calving of dunes in the simulated swarms, varying the angle of the secondary mode also affects the distribution of nearest downwind neighbour lateral offsets as shown in figure 19. For $\theta_b = 36°$ we observe a distribution which looks fairly similar to the unimodal simulations except that the secondary peaks just outside $\delta y/W = \pm 1$ have disappeared. As the angular separation increases, we see that the left-hand peak (indicating that $y_{downwind} > y_{upwind}$) becomes larger than the other peak and that the depression between the two peaks becomes less pronounced, until finally the

trough is barely visible in figure 19d).





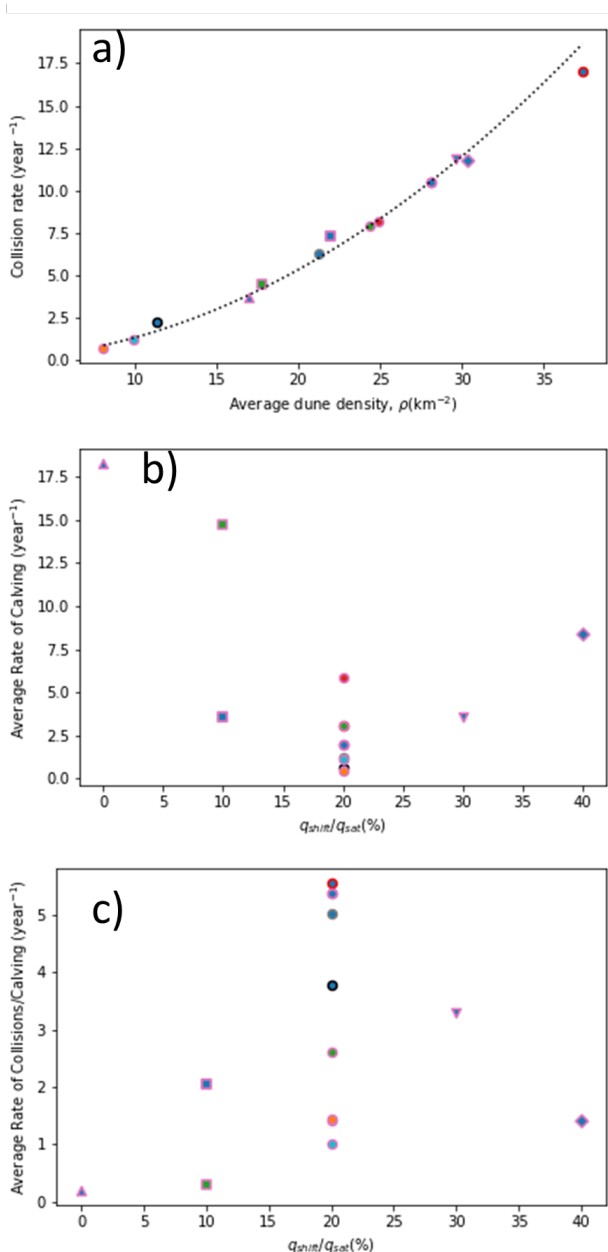

**Figure 18.** a) The average rate of collisions in each of the simulated swarms calculated as the time-averaged number of collisions per year after the swarms had stabilised. The dotted line is proportional to the square of the density. b) The time-averaged rate of calving in the simulated swarms as a function of $q-shift$ c) The time-averaged ratio of collision and calving rates across the different simulated swarms as a function of $q_{shift}$. In each of the plots $q_{shift}/q_{sat} = 0, 0.1, 0.2, 0.3$, and $0.4$ are represented respectively by upwards triangles, squares, circles, downward triangles, and diamonds. The colours of the borders of the marker are black, grey, pink, and red for respectively $\rho_0 = 12$, $24$, $37$, $61 \mathrm{km}^{-2}$. Angular separation of $\theta_b = 0°$ (i.e. unimodal), $36°$, $72°$, $108°$, $120°$ are represented by the body colours blue, green, red, cyan, and orange.



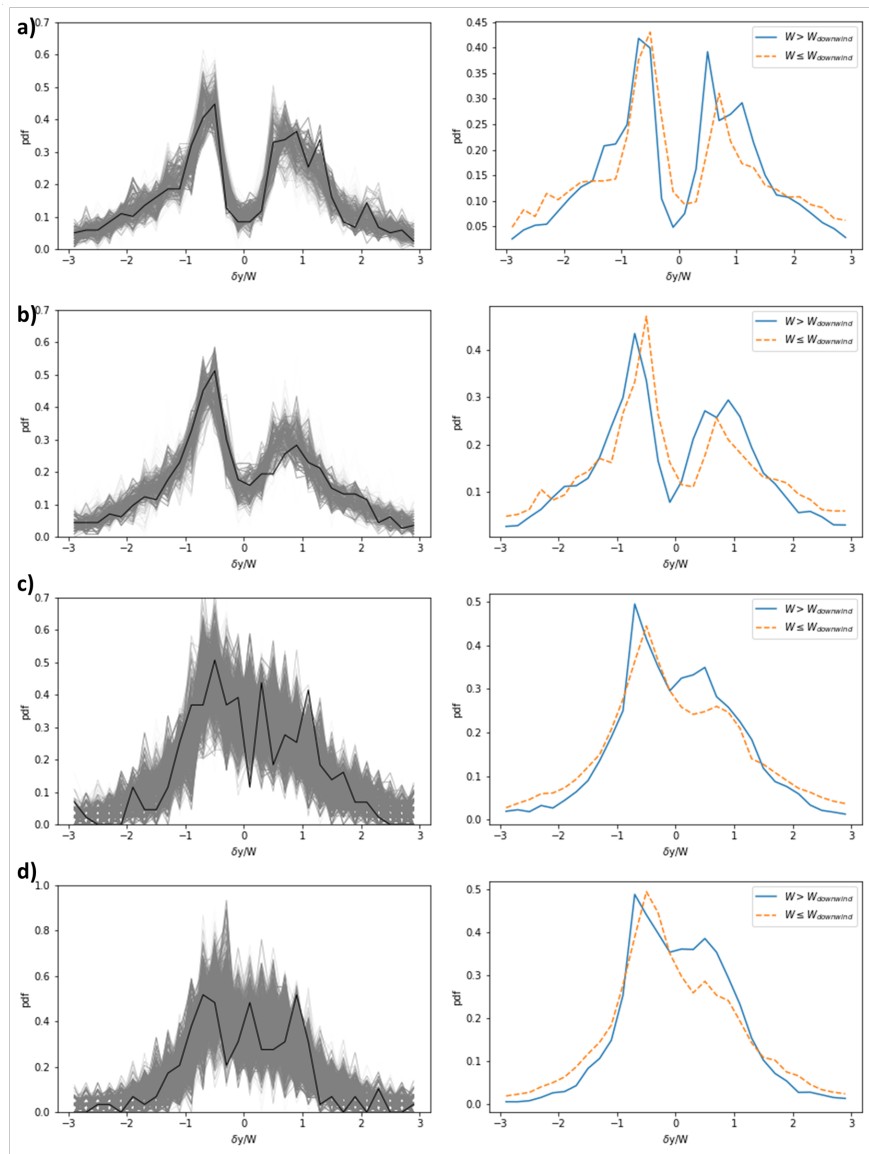

**Figure 19.** The lateral offset of the nearest downwind neighbour of dunes normalised by the width of the upwind dune for $q_{shift}/q_{sat} = 0.2$ and $\theta_b =36°$, $72°$, $108°$, and $120°$ in a)-d) respectively. The figures on the left shown the distributions each year with the most recent years shown in the darker colours. On the right, are the time averaged distributions for the two cases when the upwind dune is smaller or larger than the downwind neighbour.

## 3 Discussion

The efficiency of the TFABM has meant that we have been able to perform a significant number of swarmscale simulations and investigate the impact of a number of parameters on the types of swarms that are produced. Perhaps the most important result



in the context of barchan ABMs is that we are able to produce swarms where the average dune size stabilises with downwind

distances for unidirectional simulations and injection densities which are relatively low ($\lesssim 61 \text{km}^{-2}$) when $q_{shift}/q_{sat} \gtrsim 0.2$. In contrast, previous barchan ABMs (Lima et al., 2002; Durán et al., 2011; Worman et al., 2013) have not been able to produce longitudinally stable dune sizes, or else produce size distributions that are very different to those observed for real-world swarms (Worman et al., 2013; Génois et al., 2013a). We find that in our simulated swarms the collision dynamics stabilise with approximately $66-70\%$ of collisions resulting in fragmentation, $20-25\%$ in material exchange, and $8-10\%$ in merging. This

conforms to a previous suggestion that fragmentation collisions are necessary for longitudinal stability of the size distribution (Durán et al., 2011).

We find that the transition to fragmentation-dominated collisions occurs in the range $0.1q_{sat} \lesssim q_{shift} \lesssim 0.2q_{sat}$, while the widths of real-world asymmetry distributions (Robson et al., 2022) correspond to the same range of $q_{shift}$. Furthermore, it has

been shown (Robson and Baas, 2023) that similar values of $q_{shift}$ are able to reproduce many of the collisions from water-tank experiments (Assis and Franklin, 2020) and cellular automata (Katsuki et al., 2005). Thus it appears that $q_{shift} \sim 0.2q_{sat}$ is consistent with longitudinally stable dune sizes, fragmentation-dominated collisions, the spread of bilateral asymmetry distributions, and collisions observed using microscopic models and subaqueous experiments (Robson and Baas, 2023). In the absence of direct measurements of the rate at which asymmetric dunes revert to a symmetric morphology (Parteli et al., 2014a),

this provides at least a strong constraint on the value of $q_{shift}$. Interestingly, this range of $q_{shift}$ is also consistent with the range of the intensity of isotropic turbulence in atmospheric boundary layers (Kaimal and Finnigan, 1994).

It has been suggested using mean-field modelling that collisions occur significantly more frequently than calving events (Robson et al., 2022), however, we find that the relative rates of these processes varied considerably amongst the different sim-

ulations we performed. Due to the collision rule of the TFABM, calving can be triggered in the iteration following a collision (Robson and Baas, 2023) making it difficult to unpick the relative frequencies of the events since we cannot separate the calving events due to collisions from those that occurred on their own. Although we cannot directly compare our results to the previously reported probabilities (Robson et al., 2022), we can however report that, in the unimodal simulations, collisions occurred more frequently in all swarms except when $q_{shift} = 0$. The maximum relative frequency occurred when $q_{shift} = 0.2q_{sat}$ for

which collisions occurred between 3.3 and 5 times as often as calving depending on the density. These relative rates are still much lower than in the mean-field modelling (Robson et al., 2022) however, since it is possible that many of the observed calving events occurred because of a collision, we cannot say whether or not our results differ significantly from the previous findings. Nevertheless, the observation that $q_{shift} = 0.2q_{sat}$ corresponds to the previously reported domination of collisions over calving (Robson et al., 2022) adds further weight to the constraint on $q_{shift}$. In all cases, the relative rate of calving was

higher in the bimodal runs compared to the unimodal ones which likely results from the increased asymmetry of dunes due to the wind variation.



Although we have been successful in producing size distributions which are homogeneous with downwind distance, in most simulations we find that the density of dunes decreases with distance. This coarsening was also seen in previous agent-based

models (Lima et al., 2002; Worman et al., 2013; Durán et al., 2011). It seems likely that the cause of this is that the free flux entering at the upwind boundary of the swarm is absorbed by the dunes in the upwind portion of the swarm, so that the only flux propagating through the rest of the swarm is that which streams off the horns of those upwind dunes. In our model this flux propagates downwind only in narrow channels behind the horns of the upwind dunes such that downwind dunes only persist if their lateral position coincides with this streaming flux. This effect is what produces the strong peaks in the nearest

downwind neighbour lateral offset distributions, while the flux shadow cast by the bulk of the dune leads to the trough at the centre of the distributions (see figure 5). Although this effect is also seen for real-world swarms (see figure 6) when the upwind dune is larger than its downwind neighbour, in the case where the downwind dune is wider, the trough in the centre of the distribution is less pronounced or entirely absent (Elbelrhiti et al., 2008). It seems likely, therefore, that the ability of large dunes to align with the centre of small ones is necessary for the dune density to remain constant with downwind distance. It

may be necessary to extend the model to include a lateral diffusion term in the propagation of sand flux (Lima et al., 2002). We do observe longitudinal stability of dune density when $\rho_0 = 12\text{km}^{-2}$ i.e. when the dune density was low. This density is still consistent with sparse swarms such as in La Joya, Peru (Elbelrhiti et al., 2008) or the sparsest swarms in Tarfaya, which have a dune density of only $\sim 20\text{km}^{-2}$ (Elbelrhiti et al., 2008), but is well below the density of other swarms, such as those measured in Tarfaya and Mauritania in (Robson et al., 2022) which have a density of around 200-250km$^{-2}$(Elbelrhiti et al., 2008).


The fact that density in sparse swarms can be stable (see figure 9 a) using the TFABM model supports the hypothesis that it is the redistribution of the available flux from a wide spread to confinement in narrow channels that leads to the decrease in dune density, since with a lower density at the upwind portion of the swarm, the free flux supplied at the upwind boundary can propagate further into the system. Diffusion of flux is likely to be more important in denser swarms as well, since the airflow

will be perturbed around dunes (Schwämmle and Herrmann, 2005), leading to a spread in the direction of flux propagation. It is also worth noting that in the sparse swarm the average size of dunes is larger, so that each dune may be able to align with multiple horn outfluxes, something which is less likely for smaller dunes.

Recently, it has been suggested that when subject to a change in boundary conditions, a dune field will experience a higher

rate of interactions during the transition before reaching a new stable state consistent with the changed conditions (Marvin et al., 2023). We tested this hypothesis by altering the injection density (i.e. external sand supply) to a previously stabilised swarm. Although we see that the swarm transitions to a new state after the change in boundary conditions, and that this transition involves a wave-like propagation of the new properties through the swarm (see figure 13), we do not see a sharp increase in the rate of collisions in this transition itself. Instead, we observe a smooth asymptotic adjustment in the rate of collisions to

that of the new equilibrium. Instead, the main control on collision rate is the local density of dunes, reflected in the fact that swarms with a higher overall density experience a higher rate of collisions even though an identical $q_{shift}$ ensures consistent collision behaviour. The dependence of collision rate on density is also evident in the average location of collisions, which is



strongly affected by the shape of the longitudinal dune density profile.

A key novelty of our study compared to previous barchan ABMs is the ability to simulate swarms under bimodal wind regimes. We find that, depending on the angular separation of the wind directions, the resulting swarms are very different. The most notable difference being that, for obtuse angular separation, the swarms comprise primarily a narrow longitudinal stream of dunes rather than occupying the entire lateral dimension of the domain. Dunes outside of this narrow stream grow very large compared to those in the stream. This suggests that so called "corridor" formation that has been reported in real-world swarms, where dunes in different longitudinal strips within a single swarm may have very different sizes (Elbelrhiti et al., 2008), may form as a result of obtuse bimodality in the wind regime.

Based on the simulation of isolated dunes using the TFABM (Robson and Baas, 2023) and microscopic simulations of asymmetric barchans (Parteli et al., 2014a; Lv et al., 2016), we had expected to observe a transition from Bagnold-like (Bagnold, 1941) asymmetry growth to Tsoar-model growth (Tsoar, 1984; Tsoar and Parteli, 2016), as the secondary mode changes from an acute to an obtuse angle of separation. Instead, although we observe that increasingly acute angles cause dunes to become more asymmetric and, consequently, calving more, we find that for obtuse wind mode separations the type of asymmetry observed remains the same, although less dramatic. This demonstrates that dunes within a swarm behave differently to isolated dunes, which may explain why studies into the origin of asymmetry in real-world swarms have struggled to find conclusive evidence in favour of either the Bagnold-model or Tsoar-model growth (Bourke, 2010).

It is likely that the way that the dunes are interacting via the sand flux field is producing Bagnold-model growth even under an obtuse secondary mode. Specifically, it may be that the alignment of the dunes - with barchans preferentially aligning with the port flank of their upwind neighbours - means that under the primary wind mode they might receive more flux on their starboard flank counteracting any growth of the port flank under the secondary mode. This may explain why dunes are, on average, less asymmetric in the obtuse secondary mode simulations. For acute angular separations, on the other hand, dunes still typically align with the port flank of their upwind neighbour, meaning that during both the primary and secondary modes the starboard flank receives more flux producing greater levels of asymmetry.

The fact that most dunes display the same manner of asymmetry (larger starboard flank) in the bimodal simulations may explain why merging collisions occur more frequently and over a wider range of lateral offsets in the bimodal runs compared with the unimodal simulations. When interacting dunes have the same asymmetry orientation a collision involves the larger flank of one dune intersecting with the smaller flank of the other dune. This is more likely to result in merging or exchange rather than fragmentation since the combined volume of the merged small and large flank will be comparable to the volume of the larger flank itself, thus it is unlikely that the collision algorithm of the TFABM will produce a fragmentation.



# 4 Conclusions

We have demonstrated that the Two-Flank Agent-Based model is capable of generating swarms of barchans which have longitudinally homogeneous size distributions which had not been produced using previous agent-based models. For sparse swarms, the dune number density is also longitudinally homogeneous as expected for real-world swarms, however, in most cases, the density of dunes decreases with downwind distance suggesting that additional effects such as the lateral diffusion of horn flux must be included to produce realistic dense swarms.

Unlike with previous agent-based models the Two-Flank Agent-Based model is capable of simulating swarms under bimodal winds which revealed that bimodality may be responsible for asymmetry distributions centred away from unity, and also asymmetries in the alignment of dunes with their downwind neighbours. We demonstrated that bimodal winds impact the rate of calving and the form of collisions, generally leading to more merging and exchange collisions and fewer fragmentation collisions. We demonstrated, also, that Bagnold-model asymmetry growth appears to dominate whereas Tsoar-model growth is not seen, likely due to the interaction of the dunes through sand flux.

Future works may seek to investigate the effects of other model parameters and wind regimes in which the duration and strength of oblique winds are varied as well as the angles.

*Code and data availability.* Two-Flank Agent-Based model has been developed openly and version 2 is published at Robson, D. (2023). DTRobson/TwoFlankABMModel: TwoFlankBarchanABMv2. Zenodo. https://doi.org/10.5281/zenodo.10252816 it is being developed at https://github.com/DTRobson/TwoFlankABMModel/releases/tag/ABM. Animations showing the swarm simulations using the model are available at Robson, D. (2023, December 3). Barchan Swarm Simulations Using the Two-Flank Agent-Based Model. Zenodo. https://doi.org/10.5281/zenodo.10252764

*Author contributions.* DT Robson led the development of the model, performed the simulations described, and wrote the initial draft of this manuscript. ACW Baas assisted in the design of the model and the research strategy and carried out text edits on the submitted manuscript.

*Competing interests.* Andreas Baas is a member of the editorial board of journal Earth Surface Dynamics. The authors have no other competing interests to declare.

*Acknowledgements.* DT Robson was supported by the EPSRC Centre for Doctoral Training in Cross-Disciplinary Approaches to Non-Equilibrium Systems (CANES EP/L015854/1).



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
