# Peer review of "Barchan swarm dynamics from a Two-Flank Agent-Based Model"

_EGUsphere, 2023_

## Author Response (AR1)

Dear editor and reviewers,

We would like to thank you for the time you have taken to look through and review our original manuscript. There were many suggestion from the reviewers which we have taken on board and which have benefited the revised manuscript.

In particular it was felt that in the original we focused too heavily on the model and not enough on the implications of our findings. We feel we have addressed that concern and now much more clearly relate the simulation results to real-world data.

As suggested by one reviewer, we have also included new simulations using a different rule for the outflux from dunes now comparing directly the results from the two different approaches. We believe the revised manuscript is improved from the original and we look forward to receiving comments.

Below are responses to the specific comments from the reviewers.

**Reviewer 1**

Dear editors and authors,

Thank you for giving me this opportunity to learn and review.

The authors simulated the dynamic processes of Barchan swarm using the TFAB model, Firstly, it is undeniable that the authors have attempted to advance our understanding of the dynamic processes of Barchan swarm, which is a very interesting study. Meanwhile, the manuscript is generally well-written, and the model clearly communicated. However, the main results of the model are not well presented in the manuscript, and The simulation results have significant limitations compared to real-world conditions. Thus, I think the manuscript will be improved with a deeper consideration and discussion of some topics that I detail below.

Line 1-5: Although I'm pleased to see this article, I was not very satisfied when I read the abstract because you did not highlight your core points. Firstly, you merely inform everyone that you have undertaken such a work, but you do not explain why it is necessary to do it, or what new and interesting contributions it can offer. Secondly, when you state "unlike previous agent-based models, we are able to..." it tends to emphasize the contributions of the model too heavily, in fact, the innovations regarding the model have already been published (e.g. Robson, D 2023; GRL).

We have rewritten the abstract to make clearer why the findings of this work are important steps forward in our understanding of the dynamics of barchan swarms.

Line 54-57: Firstly, your work is meaningful, it reports the dynamic processes of barchan swarms under unidirectional as well as bimodal winds. However, the content fails to capture the reader's interest because it overly emphasizes the importance of your model, thereby neglecting the insights this model could provide into the dynamic

mechanisms of barchan swarms. Second, you said that your model can also simulate the formation of asymmetric bedforms under the influence of variable wind regimes. Is this simulation result accurate? Can it be verified in the field? I believe that there is a significant difference between the simulation and observed results in the field, at least, it cannot represent all cases.

Thank you for acknowledging the meaning of our work and for pointing out that we do not emphasise enough its relevance to a general reader, we have addressed this in our revisions throughout the manuscript especially in the abstract, line 34 of the introduction and section 3.1.3.

On the second point, we have previously shown in Robson and Baas *GRL* (2023) that our model can reproduce asymmetry growth similar to the Bagnold model in the case of bimodal winds with acute angular separation and similar to the Tsoar model in the case of obtuse angular separation. The same dependence of the growth mode on the angular distance between modes of wind regime has previously been shown in Parteli et al. *Aeolian Research* (2014) and Lv et al. *Environmental Earth Sciences* (2016). For acute angular separation of modes, our results are very similar to Parteli et al and we will include this in supplementary material during the revision. We have also observed very similar behaviour in real world barchan swarms, especially south of Akjoujt, Mauritania in which the wind regime displays two distinct modes separated by a large acute angle.

However, we do accept that the growth of asymmetry according to the Tsoar model with obtuse angular separation is somewhat less pronounced that in the other works, this is because the asymmetry observed to arise with obtuse separation in the other simulations is the significant longitudinal extension of one of the horns (and eventually a transition to a seif or linear dune), our model can capture some of the initial stages of such an asymmetry growth, but is not able to capture the extreme extension of the horns after some time. We have made this clear in the revised manuscript from line 91. Since we now do not include obtuse separation simulations in the revised manuscript this is not so much of a problem as in the original version

Line 202-204: Although you have accounted for the distribution of dune asymmetry, I found that the simulation may not reflect the morphological characteristics of barchan swarm in the real world.

We have now included in figures 6 and 13 the asymmetry distributions of real-world barchan swarms alongside those of the simulated swarms to allow for a more direct comparison.

Line 205-207: Can you provide some photos? I would like to see real, similar evidence, which might be more convincing.

We have now included in figures 6 and 13 the asymmetry distributions of real-world barchan swarms alongside those of the simulated swarms to allow for a more direct comparison.

Line 295: Why did you choose these specific angles (36°, 72°, 108°, and 120°)? Do you believe that a 73-degree angle will still lead to the formation of Barchan swarm in the real world? Can you give me an example?

The angles were initially chosen to be 36, 72, 108, 144, 180 i.e. to divide the angular space evenly (the other 180 degrees would yield mirror image results). However, after receiving the reviews, we have now focused on acute angular separation only and now consider on 22.5, 45, and 67.5 degree separations. Comparing these to realistic swarms, the wind rose for the Mauritanian dunes which we describe several times in the revised manuscript can by found in figure 5 of Robson and Baas *ESPL* (2024) exhibiting and angular separation of ~60degrees.

Line 289-355 : For the result of bimodal simulations, I think they need to be discussed. First of all, under bidirectional wind conditions, it is difficult for barchan dunes to reach or maintain a stable state. Secondly, with wind at an angle, the horns of crescent-shaped dunes are very asymmetrical. I noticed that in Figure 14, the horns of many dunes are nearly symmetrical in size, which clearly is not reasonable. Lastly, under multi-directional wind conditions, transverse sand ridges would form on the windward side facing the airflow, but I found that your simulation results did not show this; instead, they more closely resemble small chains of crescent-shaped dunes.

Yes, our model is an agent-based model so the only types of dunes that can form are the agents which we have defined – so far this is only barchan dunes so transverse ridges cannot occur. The dunes in our simulations under bimodal wind do move away from a typical asymmetric shape as shown in the revised manuscript in figure 13 but we do not reach the levels of asymmetry seen in some locations. We discuss this limitation in lines 353 and 442 of the revised manuscript.

Line 450: For the angular separation of the wind direction, the asymmetry between the left and right horns of the Barchan dunes should be significant, but in your simulation results, I see that it is almost symmetrical.

Same as above, we show a comparison with real-world swarms in figure 13 and discuss this point in lines 353 and 442.

Figures and References: The visuals of Figures in the manuscript should be enhanced to make them more appealing. While your simulation results are commendable, previous research should not be overlooked; thus, it may be necessary to reference some papers on field observations and wind tunnel simulations in your manuscript (e.g. Cai et al., 2021 CATENA).

We have not sought to overlook any research in this work and have included additional citations since the original manuscript. We now compare all results directly to field observations of barchan swarms. We also discuss how results such as Cai et al. *Catena* (2021) might be included into the model on line 415.

**Reviewer 2**

The manuscript "Barchan swarm dynamics from a two-flank agent-based model" presents simulations of barchan fields (or swarm) under different wind and boundary condition using a simplified model of dune interactions. Although the collective dynamics of dune fields is a relevant and interesting problem, the authors didn't properly validate the agent-based model before moving on to complex scenarios. Of course, a simple exploratory-type model can sometimes be very valuable in shedding light on a complex problem. However, the authors don't emphasize this aspect sufficiently and we get the impression that this is a solved problem. In particular, there is no discussion of the many approximations and assumptions underlying their approach, and how their work could be improved using physical models.

Thank you for your time in reading our manuscript we now make some of the assumptions more explicit (e.g. lines 89 and 114). Such assumptions are necessary to simplify the dynamics of large systems such as barchan swarms which are untractable for more involved models. We do not agree that the model has not been properly validated, in Robson and Baas *GRL* (2023) we have already described the phenomenology of our model for many simple cases and how this compares to other types of modelling as well as observations. In particular, in the supplementary material of that work (referred to in the paper), we provide an analysis comparing the relative success of all existing barchan agent-based models at reproducing collisions from both water-tank and cellular automaton simulations.

As pointed out in detail below, the manuscript can benefit from a more in-depth discussion of the assumptions and approximations, as well as existing literature on physical dune models. In addition, the fact that mass might not be conserved in their model should be addressed in detail.

Mass is conserved in the model, this will be discussed further in other comments.

Regarding the dune model:

1. Section 2.2. The dune mass balance equation (Eq.2) is not validated neither here nor in the original paper (Robson and Bass, 2023). I know this is a published model, but a lack of validation with either data or physical models has to be discussed. For example, the physical model in Duran et al. ESPL (2010) clearly shows that the outflux from a

barchan horn is function of the influx (perhaps because the horn width depends on the influx). In contrast, in the model presented by the authors the outflux is just proportional to the saturated flux q_sat.

The dune mass balance equation used in this model is a well-known expression which we did not create, but was first suggested in Hersen et al *Phys. Rev. Lett.* (2004) based upon numerical simulations and has subsequently been used in the agent-based model of Worman et al. *Geology* (2013).  However, we accept that the scaling of influx and outflux reported in Duran et al. *ESPL* (2010) is also well-known.  In fact, we had already included the possibility of setting the outflux using the relation from that work in the code of our model.   We now introduce the influx scaled outflux from Duran et al. *ESPL* (2010) in line 109 and compare the swarms unscaled flux (from the original manuscript) The comparison between the two is now a major feature of the revised manuscript.

2. Section 2.2. The central parameter of the model (q_shift) is completely phenomenological and unconstrained -- btw this would have been the perfect application of physical models simulating the evolution of asymmetrical barchans in unidirectional winds. This parameter is introduced to simulate a lateral transfer of mass between the two flanks. However, this cannot be the case because there is no mass exchange between the two sides of a barchan (from symmetry, the transversal flux q_y at the central dune slice is 0). In fact, the arms become more equal because of an imbalance between the sand captured and emitted (q_out and q_in). The somewhat arbitrary nature of this parameter and its implication should be addressed in detail when introducing the model and also at the end of the introduction and the discussion section. This would help the reader differentiate between physical and phenomenological aspects of the model and thus the results.

We agree the *q_shift* is an unconstrained parameter describing a phenomenological process within the model.  However, we believe that the fact that a small range of the parameter is able to explain a wide range of observed properties of both collisions and swarm-scale properties suggests that it may be close to representing a physical process in reality, this is something we aim to explore in the future.  We do not agree that the flux at the central dune slice is 0, particularly since this assertion is, as you say, based upon the symmetry.  Thus, we see no reason to prevent this from being non-zero in the case of asymmetric dunes.  Furthermore, as we have said, this parameter pertains not only to flux on the surface of the dune, but also (probably more importantly) to internal restructuring of the barchan shape (e.g. if material accumulates on one side rather than the other then there may be some avalanching in the transverse direction from one flank to the other).  Such processes have not been well-understood thus far, and so we are unable to rely on existing physical models.  We have, however, as the reviewer suggested made it more clear in the revised manuscript that this is a phenomenological parameter which relates to a physical process that is not yet

properly modelled.  This is done at lines 114 where we also provide a more general interpretation of the parameter and the process which it describes.

3. Sections 2.3 and 2.4. The outcome of calving and collisions is not validated (not even in the original 2023 paper). The authors could at least compare it to the physical simulations of dune collisions in Duran et al. 2005, 2009 and 2011, and existing water dune experiments.

Validation of the outcomes of collisions from simulations and water experiments was in fact provided in the supplementary material of the original 2023 paper (and mentioned in that paper and referenced in this manuscript at line 175 and 436) and is available open access at the following link https://agupubs.onlinelibrary.wiley.com/doi/full/10.1029/2023GL105182.  Spontaneous calving is infrequent in all of the swarm simulations and previous mean-field modelling (Robson et al. *Physica A* (2022)) has suggested that it is not necessary to produce the size distributions observed for real-world swarms.

4. In particular, the spontaneous calving of large asymmetric dunes is not obvious at all. Following Eq.(2), the dune would become symmetric again after a given time. The authors should discuss the effects of their assumptions in the model outcome, e.g. what would happen if calving is ignored/suppressed?

Barchan dunes are thought to become symmetric after a given time, if this is not the case then any perturbation away from the symmetric morphology would persist and we would not see asymmetry distributions centred around 1 even in the case of unimodal wind.  See figure 3f in Assis and Franklin *GRL* (2020) which shows a decrease in asymmetry with time and also discussions in Parteli et al. *Aeolian Research* (2014). Calving is also rare in all of the simulations we performed.  We discuss this at several places in the revised text including line 120.

5. What is "fragmentation" in Fig.2? I couldn't find a definition. Similarly with "exchange". Also, why there is no fragmentation when q_shift = 0?

We now define fragmentation explicitly from line 172.  Fragmentation doesn't occur (except rarely) when q_shift = 0 since with that parameter even highly asymmetric dune are stable and do not break apart very often.  This is discussed in detail in the original paper in which the model was introduced.

6. The authors should include a figure defining the different parts of the barchan and also the different outcomes from collisions. I know some of this was already included in their previous paper, but it is very difficult to follow this one without some visual guide.

The terms which we use in this manuscript are well-established terminology for the morphology of barchans.  We have provided additional description of several of the

processes in the model (line 172) which we hope make it easier to follow.  However, since the manuscript is already quite long we have not included an additional figure.

Regarding the results:

7. One thing I found very strange was the spatial decrease of dune density for a constant average dune width (Figs. 3 and 9). I mean, if the average dune width (and thus volume) is constant spatially, then a decrease in the number of dunes suggest sand is not conserved. The authors must check that their equations conserve the mass. For example, in Duran et al. (2011), the cover fraction of dunes (dune area/total area) was constant downwind, which is consistent with mass conservation.

Preservation of area in Duran et al. (2011) is not consistent with mass conservation as asserted here.  We have checked the model implementation and there can be no loss of sediment as a whole.  However, since the system is open (i.e. no periodic boundaries) any sediment which ceases to be in dunes (i.e. when a dune is lost due to becoming too small) is converted into sand flux which effectively propagates "infinitely" fast and so exits the simulation space unless captured by a dune.  There is no reason to expect that real-world swarms are closed systems (in fact this is certainly not the case).  This balance between sediment contained within dunes and in free flux is inherent to any model (including Duran et al. (2011)).

8. Furthermore, even if mass is somehow conserved in their model, the authors should discuss the meaning of a dune field model where dune density continuously decrease, which implies dunes eventually disappear downwind.

We discuss this now at several places in the text.  Note that what the reviewer suggests is in fact precisely what Bagnold (1941) asserted was the behaviour of real-world systems of barchans in paragraph 8 of chapter 14 (page 218 of the 1954 reprint) e.g. "[it is] common for barchan belts to terminate because the frequency of their occurrence dwindles to nothing".

9. It would be interesting to calculate the PDF of dune width. In one of their previous papers they presented a mean-field model that captured real distributions using a similar set of collision outcomes. How that compares with the explicit simulations shown here?

We now show a stable PDF of dune width and that it can be described by a log-normal distribution in figure 10 of the revised manuscript.

10. Finally, the authors should discuss the pros and cons of the different approaches. For example, Duran et al. (2011) simulated dune fields by leveraging the results of physical dune simulations but reducing the outcome of collisions to only two dunes (for simplicity), which clearly wasn't enough to capture the spatial uniformity. Here, on the other hand, the authors introduce purely phenomenological relations and are able to

reproduce the uniformity of real fields for some conditions. This should help clarifying the open issues and next steps within the topic.

We now discuss throughout the revised manuscript questions raised by this work and its overall implications for understanding barchan swarms. In particular we have rewritten the entire conclusion and most of the discussion as well as section 3.1.1 how the findings of this work highlight open issues and future opportunities.

**Comment 1**

"We thoroughly enjoyed reading your preprint and are excited about the prospect of using your model to explore dune swarm dynamics."

Thank you, we look forward to seeing what can be done in the future!

"We would like to take the opportunity this format offers to address a point you raise in your manuscript. Specifically, you state that shifting upwind boundary conditions does not result in a transient/local increase in dune-interaction density, but rather to a gradual relaxation to a new steady state, which you contend is contrary to the results of Marvin et al. (2023). Here, we argue that model results appear, in fact, to support the findings of Marvin et al. (2023).

First and foremost, the comparison is not of apples to apples. Although your simulations run long enough for dune width and number to stabilize, small, isolated dunes that are not in equilibrium with the system are constantly fed at the upwind boundary. Thus, when averaged over the whole domain, dune pattern statistics reflect some average over the whole pattern transition, from disequilibrium, to adjusting, to equilibrated. In contrast, Marvin et al. (2023) do not consider field-averaged pattern statistics, but how pattern statistics would vary spatially downwind. Their results suggest that dune-interaction density, under the conditions of your simulation, should first increase then decrease in the downwind direction (similar to the example in their Figure 3E) – once the whole dune field is formed and pattern statistics have equilibrated downwind. Although it is difficult for us to assess quantitatively whether it is the case, your model does seem to produce a local increase in dune interactions around 2-5 km (e.g., your Figure 1).

This subtle misrepresentation of the Marvin et al. (2023) findings is reflected in a few places in the preprint:"

"- At line 258-262: you state that the portion of the dune field between 0-3 km is where the greatest change in density occurs, resulting from the change in boundary conditions, and that according to Marvin et al. (2023) it is also where we should expect the highest rate of collisions. However, the expectation from Marvin et al. (2023) would be for dune density to first increase so dunes can start exchanging sand; only at that

point (in space) would the spatial density of dune interactions start increasing. In fact, your finding that dune-interaction density is highest where dune density is highest is consistent with the finding of Marvin et al. (2023) that the background level of dune interactions increases with decreasing dune spacing."

"- At line 283-286: the observed gradual relaxation in average number of collisions with time is not inconsistent with the results of Marvin et al. (2023). The results of Marvin et al. (2023) suggest that, once your domain is full of dunes and the number of dunes in the domain has plateaued, the spatial density of dune interactions should increase and then decrease downwind, with the local enhancement in dune-interaction density reflecting the pattern's response to upwind changes in boundary conditions. In your model, you feed small, isolated dunes, which then interact downwind to, over time, start forming an equilibrated swarm further downwind. Thus, at early model epochs, the domain only contains small, isolated dunes (low number of interactions). As time progresses, those dunes migrate downwind and start interacting, increasing the number of interactions present in the whole domain, until they reach some equilibrated pattern, at which point the dunes migrate downwind without changing pattern statistics anymore. At that point in time, pattern statistics averaged over the whole domain have plateaued, but a spatial increase and then decrease in dune-interaction density in the downwind direction could still (and in fact does seem to) exist, at least qualitatively consistent with Marvin et al. (2023)."

"- At line 439-447: Again here, your statements compare the time evolution of domain-averaged dune pattern statistics to results that instead pertain to the spatial evolution of local dune pattern statistics at a given point in time. This difference in what is actually measured in the two studies seems to reconcile their results."

It is indeed true that the results we presented in the original manuscript relate to spatial averaged statistics which is not what was reported in your work. In the revised manuscript (line 313) we have rewritten the comparison between our results and those of Marvin et al. (2023). In particular, we describe the difficulty in observing a change in local collision rate during the boundary condition change because instantaneous local interaction rate is highly variable in the swarms as a whole. Therefore we can only easily measure long-time averages of local interaction rates. On the other hand, the wavefront of changing boundary conditions propagates through the swarm rapidly meaning that any short-lived increase in local interaction rate would be masked in a long-time average.

"Second, we note an even subtler difference that might result in qualitatively but not quantitatively similar results between the two studies. All collisions are interactions, but not all interactions are collisions. Marvin et al. (2023) adopt the definition of Day and Kocurek (2018) for a dune interaction, i.e., any location where two adjacent dune crestlines approach each other within 10% of the field-average dune spacing is counted

as a dune interaction. Whereas somewhat arbitrary, this threshold is used as a proxy to define where crestlines are close enough for dunes to exchange sand with one another. This is in contrast with your quantification of collisions, which you define as any point where a dune center of mass (CoM) impinges upon the footprint of another dune."

We definitely agree that not all interactions are collisions. Unfortunately however, it would be much more difficult to determine the density of long-range interactions such as through flux exchange. There is also something of an issue in defining where long-range interactions take place since they are, by definition, non-local. Thus if two dunes were interacting over a distance then one could imagine a number of ways of defining "where" that interaction was taking place e.g. at the upwind dune, at the downwind dune, the average position of the interacting dunes etc. It is not obvious which of these conventions would be the best to use but it is clear that the choice would have a significant impact on any measure of local interaction density.

"In summary, we believe that your results are at least qualitatively consistent with those of Marvin et al. (2023), contrary to what is currently stated in the preprint. Showing this would require considering spatial changes in interaction density once dune number in the domain has equilibrated rather than the time evolution of domain-averaged pattern statistics. We suspect that such an analysis would reconcile both studies qualitatively. A quantitative comparison would further require the consideration of dune interactions (per Day and Kocurek, 2018) rather than just collisions.

Again, we are excited about what your model will teach us about dune swarm dynamics and are thankful for your consideration of our recent results in your manuscript!

Best regards,

M. Colin Marvin and Mathieu Lapôtre"

Thank you again for the time you have taken to read our work, we look forward to seeing how we can continue to progress our understanding of dune-systems together.